# Intelligent surgical workflow recognition for endoscopic submucosal dissection with real-time animal study

Jianfeng Cao [1], Hon-Chi Yip[2] ✉, Yueyao Chen [1], Markus Scheppach[3], Xiaobei Luo[4], Hongzheng Yang[1], Ming Kit Cheng[5], Yonghao Long [1], Yueming Jin[6], Philip Wai-Yan Chiu [7] ✉, Yeung Yam [5,7,8] ✉, Helen Mei-Ling Meng [8] ✉ & Qi Dou[1] ✉

Recent advancements in artificial intelligence have witnessed human-level performance; however, AI-enabled cognitive assistance for therapeutic procedures has not been fully explored nor pre-clinically validated. Here we propose AI-Endo, an intelligent surgical workflow recognition suit, for endoscopic submucosal dissection (ESD). Our AI-Endo is trained on high-quality ESD cases from an expert endoscopist, covering a decade time expansion and consisting of 201,026 labeled frames. The learned model demonstrates outstanding performance on validation data, including cases from relatively junior endoscopists with various skill levels, procedures conducted with different endoscopy systems and therapeutic skills, and cohorts from international multi-centers. Furthermore, we integrate our AI-Endo with the Olympus endoscopic system and validate the AI-enabled cognitive assistance system with animal studies in live ESD training sessions. Dedicated data analysis from surgical phase recognition results is summarized in an automatically generated report for skill assessment.

AI-enabled video data analytics is promising to provide cognitive assistance for various clinical needs in minimally invasive surgery[1]. Analyzing the progress of surgical workflow, i.e., recognizing which surgical step/phase is ongoing at each second, is important for the standardization and support of surgical care[2,3]. For example, in endoscopic submucosal dissection (ESD), a therapeutic approach to resect early-stage gastrointestinal (GI) cancer[4,5], the smoothness and proficiency of its dissection phase can exhibit a surgeon's skill[6–8]. Using AI to accomplish such analytical assessment has the potential to promote more efficient and standardized surgical operations[9]; however, a relevant study is still in its infancy.

With advances in computer-assisted surgery in clinical practice[10–12], intelligent surgical workflow analysis has attracted increasing attention from computer scientists and surgeons. Despite promising progress has been made[13], the way to automated surgical data analysis is still encumbered by technical challenges. A core unsolved dilemma is to balance the accuracy and efficiency of AI prediction models. On the one hand, accurate surgical workflow recognition relies on the consideration

[1]Department of Computer Science and Engineering, The Chinese University of Hong Kong, Hong Kong, China. [2]Department of Surgery, The Chinese University of Hong Kong, Hong Kong, China. [3]Internal Medicine III-Gastroenterology, University Hospital of Augsburg, Augsburg, Germany. [4]Guangdong Provincial Key Laboratory of Gastroenterology, Nanfang Hospital, Southern Medical University, Guangzhou, China. [5]Department of Mechanical and Automation Engineering, The Chinese University of Hong Kong, Hong Kong, China. [6]Department of Biomedical Engineering, National University of Singapore, Singapore, Singapore. [7]Multi-scale Medical Robotics Center and The Chinese University of Hong Kong, Hong Kong, China. [8]Centre for Perceptual and Interactive Intelligence and The Chinese University of Hong Kong, Hong Kong, China. ✉e-mail: hcyip@surgery.cuhk.edu.hk; philipchiu@surgery.cuhk.edu.hk; yyam@mae.cuhk.edu.hk; hmmeng@se.cuhk.edu.hk; qidou@cuhk.edu.hk

of rich temporal information in the video, because temporal context awareness is critical for understanding sequential actions. This requires AI models to extract long-range features from a sequence of frames[14]. Existing methods, such as 3D CNNs[15,16] and temporal convolution network[17,18], still struggle with how to effectively capture global temporal information given the expansive surgical duration. On the other hand, recognized surgical phases need to be predicted in real-time, in order to fulfill intraoperative deployment in surgery. It is challenging to achieve such a high efficiency without compressing model parameters and sacrificing model performance. Although some representative works, such as TMRNet[19] and Trans-SVNet[20], achieved promising results with versatile models, their dependence on the considerable computational resources constrains their potential for clinical application. To date, how to effectively address this dilemma for successfully deploying AI models in the operating room is still an open question.

As a pivotal role in maintaining high accuracy of phase recognition, dataset quality drives the learning process of AI models with representative samples and universal features. Different from the principles in traditional video-based action recognition[21–23], expert knowledge could impact on the modeling of operational patterns in ESD surgery[24], thus determining the applicability of the AI model to various cases in the stage of clinical deployment. Therefore, developing surgical AI models has a greater need for establishing an expert dataset that covers the changes in anatomic targets, surgical tools, and how a tool is manipulated by surgeons[25]. Standardization and expertise of the dataset can not only provide typical samples that commonly occur in ESD therapeutic procedures[26] but also facilitate future downstream analysis based on the recognition results[27]. The construction of such a dataset, however, still remains to be completed due to the scarcity of experts as well as annotation protocols[28].

Despite that surgical data science has been studied for a while[29], experimental validation of deep learning models in real-world complex scenarios and/or real-time pre-clinical settings is still extremely limited. Existing literature still lacks systematic experiments on how the developed AI models are validated given various surgeon expertise (e.g., from novices to experienced ones), long-time data expansion (i.e., surgical instruments change over time) and across surgical sites (from retrospective human data to ex vivo/in vivo animal trials). All these factors would introduce data distribution shifts and are important to be experimentally considered because they may degrade the generalizability of data-driven models. In addition, how to incorporate such automated data analysis in a way that fits into clinical workflow and fulfills clinical needs is non-trivial and unclear. In these regards, systematic experiments, even live animal studies, are necessary to experimentally verify the effectiveness of AI models for real-world clinical applications. Some works have explored possibilities to incorporate intelligent functions in applications of procedural skill assessment[30,31] and future frame prediction[32] through in-silico experiments, however, these works were limited to using surgical data analytics in an offline mode, rarely considering the efficiency of burdensome models in practice. For the advancement of the clinical value of AI models, experimental results in real-world settings are frequently suggested in smart healthcare-related guidelines. To date, there is no reported work on validating AI models in live animal preclinical settings for ESD.

In this study, we proposed a deep learning-based method (named AI-Endo) for intelligent surgical workflow recognition in ESD. To achieve accurate phase recognition and real-time clinical deployment, we introduced a cascade of feature extraction and fusion modules with the ability of spatial-temporal reasoning. As the endoscopy video streams into the framework, it could not only extract representative frame-wise features but also distill temporal relations to describe complicated surgical scenes. Furthermore, we designed the framework with a light yet compelling feature backbone and dynamic feature fusion to accommodate the trade-off regarding test efficiency.

Importantly, our model learned from high-quality data collected from an expert endoscopist (with ESD experience of over 15 years), accompanied by clearly defined surgical phase definition and exhaustive frame-wise annotation. To experimentally evaluate the performance of AI-Endo in the wild, we have extensively tested its performance on external datasets including different endoscopists, various surgical tools and skills, different endoscopy systems, and multi-center datasets. Moreover, we studied the potential usage of surgical phase recognition, by integrating AI-Endo into surgical skill training sessions at CUHK Jockey Club Minimally Invasive Surgical Skills Centre. To evaluate the computational efficiency and compatibility of AI-Endo in real-time applications, we conducted a cost-effective ex vivo animal trial using video streamed from an endoscopy system to our AI workstation. Thereafter, we designed an in vivo animal study to showcase the potential of AI-Endo in standard clinical setup. A user-friendly interface was developed that could visualize real-time recognition of surgical workflow and automatically generate a summary report for data analysis toward surgical skill assessment. This study sheds light on automated surgical workflow recognition with validation in real-time pre-clinical settings for ESD.

## Results
### Developmental dataset for model training
Forty-seven endoscopy videos with full-length ESD procedures (duration 71.28 ± 36.71 min) recorded from the Endoscopy Centre of Prince of Wales Hospital in Hong Kong were used as the training cohort. All cases were performed by an expert who has over a decade of experience in ESD. Expert procedure videos were chosen as training material as AI models treated the dataset as a gold standard, and the demonstrated endoscopic and device maneuvering skills should represent expertise for operations in safety-critical situations. The dataset covered a long period from July 2008 to March 2020. The videos were recorded using the endoscopy video processor (CV-260 and CV-290, Olympus Medical Corporations, Tokyo, Japan), with a resolution of 352 × 240 or 720 × 576 at 25 fps and a resolution of 1920 × 1080 at 50 fps (i.e., frames per second). This yielded up to 3GB file size for each single case and millions of frames in total for the overall dataset. All patients' sensitive information including ID, sex, and age was de-identified and patient consent forms were waived for the retrospective cohort. IRB has been approved by the ethics committee of The Chinese University of Hong Kong.

The included cases cover a wide variability of lesion sizes, locations (i.e., rectum, stomach and esophagus) and surgical tools (i.e., dual/isolation-tipped/triangle-tipped knife). More details about the variability of the dataset are provided in Supplementary Table 1. Although the dataset spans a long time of 12 years, for the whole period the endoscopist has already achieved the level of expertise. At the start point (year 2008) of the cohort duration, the endoscopist had conducted more than 100 ESD cases on each organ of rectum, stomach, and esophagus. According to the learning curve reported in refs. 33–35, the endoscopist can be treated as an expert because the number of conducted cases is higher than the suggested bar (80/30/30 cases of rectum/stomach/esophagus, respectively). Annotation was performed on all of the retrospective datasets for surgical phase recognition.

### Annotation protocol of ESD workflow
To annotate the developmental dataset and external validation dataset, we propose a standardized ESD annotation protocol (see Fig. 1a). Four surgical phases have been defined: (1) Marking: the periphery of the target lesion would be identified, then marking would be performed by applying multiple electrocautery marks circumferentially around 5 mm away from the lesion at 2 mm intervals; (2) Injection: submucosal elevation would be achieved by injection of a mixture of solutions containing normal saline, epinephrine, or hyaluronic acid

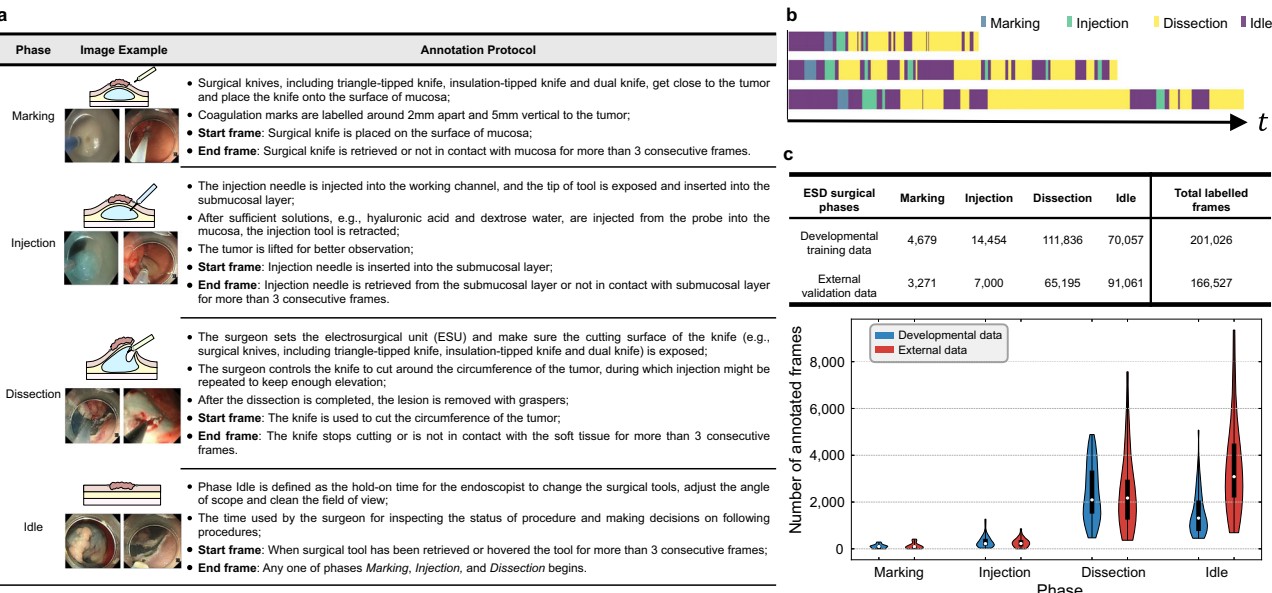

**Fig. 1 | Establishment of developmental and external datasets for ESD surgical phase recognition. a** Illustration and definition of the four surgical phases: Marking, Injection, Dissection and Idle. Examples of start and end frames are provided in Supplementary Fig. 2. **b** Phase annotation examples of surgical videos. **c** The statistical numbers of four annotated frames in our developmental training data and external validation data, and the corresponding violin plot on the distribution of annotated frames. The box indicates the median as a white point in the box and excludes the upper and lower 25% (quartiles) of data, and the whiskers extend to the extrema (Developmental data $n = 47$ cases; External data $n = 15$ cases). Source data are provided as a Source Data file.

using needle injector. Due to the difficulty in retrospective annotation and often ultra-short duration of 1–2 s, transient saline injection through the channel within the electrosurgical knives could not be separately annotated and thus would be included in the Dissection instead of Injection phase; (3) Dissection (mucosal incision and submucosal dissection): mucosa around the marking point is incised, and then submucosal layer would be dissected from the underlying muscularis propria until the target lesion is resected and removed. Hemostasis with electrosurgical knives is included because of its short duration; (4) Idle: the hold-on time spent by the endoscopists to exchange the instruments or adjust the endoscope. Each single frame is only labeled with one of the four phases, which is determined based on identifying the start and end frame of each phase as well as its temporal continuity[36].

For all the cases, we excluded frames after the tumor was completely resected. We also downsample the video to 1 fps for annotation efficiency. To ensure high-quality data, the annotation workflow consisted of three stages. First, two well-trained medical annotators independently annotated approximately 10% (5 cases, 20,446 frames) of the expert dataset based on the dataflow in Supplementary Fig. 1. The inter-rater agreement was measured using the Pearson correlation coefficient (PCC)[37], which was 0.93. This showed a high consistency of labeling between the two raters leveraging our provided annotation protocol. Supplementary Fig. 3 provides an example of the annotations from the two raters. Then, the two raters jointly labeled the whole dataset by dividing all the cohorts into approximately equal halves, with each rater individually annotating one part. After they completed all of their annotation tasks, the annotations underwent quality control by another two experienced endoscopists. The annotation assessment relied on not only visual cues but also practical experience to determine the surgical phase. Discussions happened in situations where the surgical site was highly complex or key landmarks were not clearly seen. Details on the dataflow, annotation schedule and annotation results are provided in Supplementary Note 1. Three final annotation examples (with different video durations) are shown in Fig. 1b. The number of annotated frames in the expert dataset varies across each

phase, with the phase of Dissection occupying most of the surgical time, which is also the most important and skill-demanding phase in ESD. Detailed statistics of each phase are listed in Fig. 1c. Overall, a total of 201,026 and 166,527 frames were labeled for the developmental training and external validation (described below) datasets. All the annotations in this study followed the same annotation protocol.

## External datasets for model validation

Given the complexity of anatomical scenes and the variety of procedures, it is critical to validate the applicability of the AI model to different endoscopists and operation skills. To this end, we first collected 15 cases of ESD performed in Prince of Wales Hospital in Hong Kong from April 2021 to August 2022, and 122,114 frames in total were annotated at 1 fps. These procedures were conducted by three younger endoscopists with 6, 3 and 2 years of experience in ESD respectively. Different from the developmental dataset that concentrates on data from an expert clinician with stable and proficient surgical performance, the validation data aims to reflect the variance in surgical skills in order to evaluate the model's generalizability and its potential to support skill assessment in clinical practice such as training sessions. The variation in the endoscopists' experience in the validation dataset helped to assess the AI model's tolerance to human factors that are commonly associated with different levels of expertise. This is especially important for ESD which is a typical procedure requiring a long learning curve[38].

We also conducted further validation of our AI model in ESD procedures with unseen operation techniques, i.e., the surgical skills were not present in the developmental dataset. Three ESD cases with the pocket creation method[39] and one case with line-assisted traction method[40] were acquired from Prince of Wales Hospital for the purpose. In particular, the pocket creation technique is used to improve the visualization of the dissection plane by creating a pocket in the submucosal layer after making a small mucosal incision for entry. The traction method involves using additional instruments (such as a clip with a line, snare or other commercially available traction devices) to apply counter traction. This dataset overall has 19,254 annotated

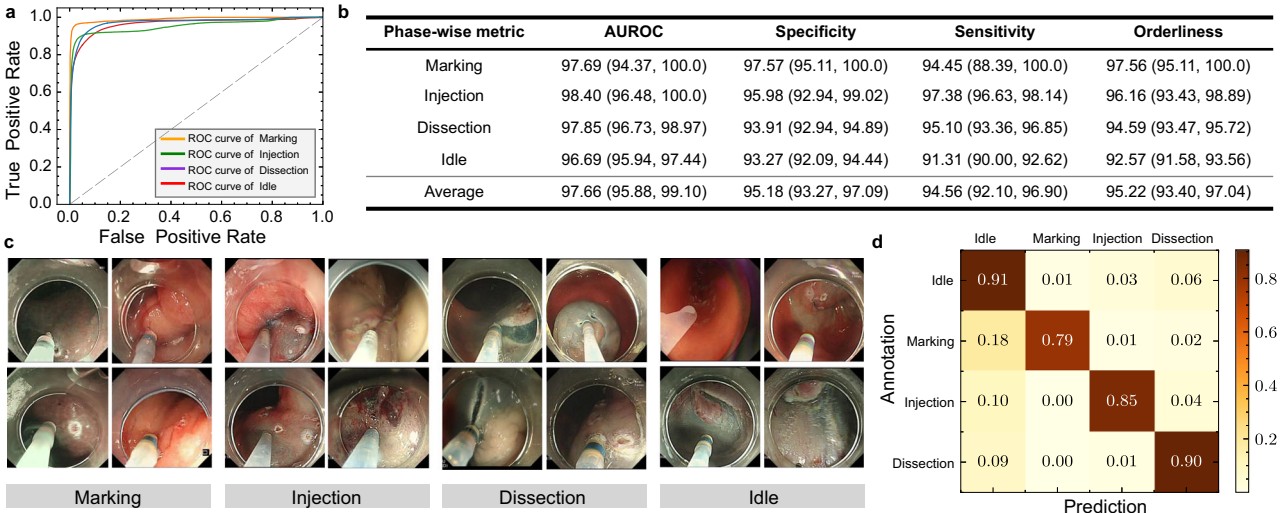

**Fig. 2 | Analysis results of 5-fold cross-validation on the developmental dataset. a** The receiver operating characteristic (ROC) curve of four phases; **b** Statistical scores of AI-Endo on four phases based on the Youden Index ($n = 47$ cases). Data are presented as 95% confidence interval; **c** Examples frames in four phases with intra-class difference and inter-class similarity; **d** The confusion matrix across four surgical phases. Source data are provided as a Source Data file.

frames. Investigation into these techniques helps observe the performance of the AI model when encountering novel styles of tool-tissue interactions.

In addition, we designed ex vivo and in vivo animal trials for the purpose of validating the integration of the AI model into the existing endoscopic system. For initial feasibility validation, we conducted four ex vivo animal trials to streamline the data flow of AI assistance into the standard ESD workflow. Afterward, we conducted in vivo experiments for validation of the whole system in real time. A total of 12 ESD procedures were performed on two live pigs in a surgical training session.

Furthermore, we collected external multi-center datasets to validate the generalizability of AI-Endo on different endoscopy systems and demographics. The first cohort contained four ESD cases from Nanfang Hospital, Southern Medical University, Guangzhou, China. This dataset was collected from a Fujifilm endoscopic system, in order to validate applicability for different imaging devices. The second cohort contained four ESD cases from Internal Medicine III-Gastroenterology, University Hospital of Augsburg, Augsburg, Germany. This dataset was collected from an Olympus endoscopic system which is the same as our developmental data, while the purpose of this external dataset is to validate the model's generalizability on patients from a different country. These two cohorts were labeled according to our annotation protocol and yielded 25,159 frames in total.

### Performance of AI-Endo model using 5-fold cross-validation on developmental dataset

For automated ESD surgical phase recognition, we propose a deep learning-based framework called AI-Endo, which inputs the video stream and embeds each frame into high-dimensional feature space. To sufficiently make use of temporal information for accurate model performance, we incorporate a cascade of feature extraction with a temporal convolution network and a global attention-based transformer to extract spatial-temporal features. Our AI-Endo is developed based on the 47 training cases in 5 folds (with sizes of 10, 10, 9, 9, 9), each one of which is used for performance evaluation while the other 4 folds are used for training the learning algorithm. Without loss of generality, this cross-validation strategy enables the developed framework to be validated on the whole developmental dataset.

The phase prediction can be obtained by taking the maximum or setting an optimal threshold on the output probabilities. Both overall and phase-wise metrics can be derived from four collections, i.e., true

positive (*TP*), true negative (*TN*), false positive (*FP*) and false negative (*FN*). For the overall performance, we adopt three commonly used criteria, i.e., average accuracy, average precision and average recall. The average accuracy ($\frac{TP+TN}{TP+TN+FP+FN}$) captures the overall ratio of correctly classified frames. The average precision ($\frac{TP}{TP+FP}$) and recall ($\frac{TP}{TP+FN}$) deliver the fraction of relevant samples in all retrieved samples and the completeness of the relevant collection. Moreover, to inspect the performance of AI-Endo on each phase, we plot the receiver operating characteristic curve (ROC) and evaluate the results of the AI inference with the area under the ROC (AUROC)[41]. Meanwhile, we refer to a summary measurement of the ROC curve, Youden Index[42,43], to apply the optimal threshold for phase prediction, yielding a set of $\hat{TP}$, $\hat{TN}$, $\hat{FP}$ and $\hat{FN}$ for each phase, which are used to calculate the specificity $\frac{\hat{TN}}{\hat{TN}+\hat{FP}}$ and sensitivity $\frac{\hat{TP}}{\hat{TP}+\hat{FN}}$ of each phase to keep coincident with the Youden Index and ROC curve. Moreover, we define the orderliness metric ($\frac{\hat{TP}+\hat{TN}}{\hat{TP}+\hat{TN}+\hat{FP}+\hat{FN}}$) for phase-wise evaluation to measure the degree of how the target frames are correctly ordered for each phase. Details on this metric are provided in Supplementary Note 2.

For evaluation results of 5-fold cross-validation on developmental dataset, our AI-Endo model obtains an average accuracy of 91.04% (CI: 89.57%, 92.51%), average precision of 88.48% (CI: 85.98%, 90.97%) and an average recall of 88.77% (CI: 85.99%, 91.54%). The high performance is attributed to the representative features learned from expert surgical videos. For the performance of AI-Endo for each phase, Fig. 2a shows the ROC curves of the four ESD phases, with specific AUROC scores as 97.69% (CI: 94.37%, 100.00%), 98.40% (CI: 96.48%, 100.00%), 97.85% (CI: 96.73%, 98.97%) and 96.69% (CI: 95.94%, 97.44%) for Marking, Injection, Dissection and Idle, respectively. In general, for all four phases, the specificity, sensitivity, and orderliness are all higher than 90% (see detailed results in Fig. 2b). This demonstrates the model's promising performance in accurately predicting ongoing surgical phases from a complex procedure. It is worth noting that the ESD surgical scenes have significant intra-class variance while considerable inter-class similarity. Figure 2c demonstrates some successfully recognized frames from each phase under such challenges. For example, in phase Dissection, the trajectory of dissection as well as the dissected surface of the submucosal layer often present variations, making phase recognition difficult. Simultaneously, the tasks of phases Marking and Injection show similarity in the interaction between the surgical tool and the surrounding tissues, such as the insert on the mucosa layer and the retraction away from the target point. Despite

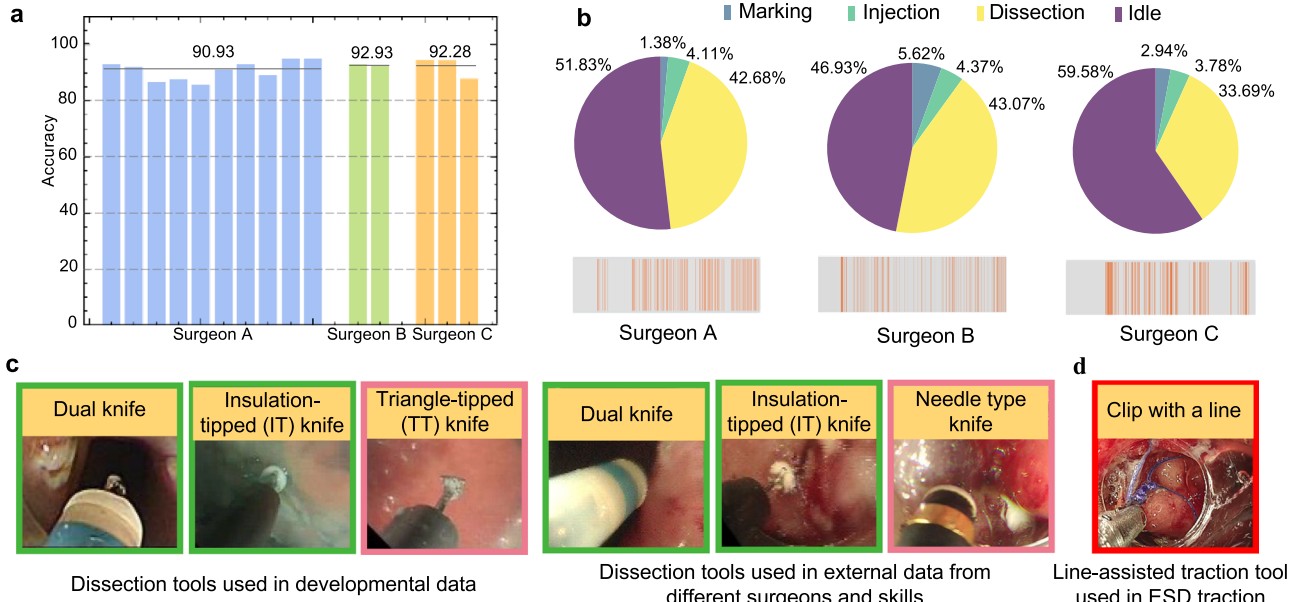

**Fig. 3 | Experimental results on validation dataset with different surgeons and skills. a** Phase recognition accuracy of AI-Endo on $n = 15$ validation ESD cases conducted by different surgeons. Each bar represents one case; **b** Proportion of phase duration and frequency of phase transition (orange-colored timestep) for surgeons A, B, and C; **c** Illustration of different dissection tools used in developmental data and external data from different surgeons and skills; **d** Line-assisted traction tool used in external data of ESD traction technique. Source data are provided as a Source Data file.

that these situations may lead to the misclassification of the AI model on similar frames from different phases (see the confusion matrix in Fig. 2d), the proposed AI-Endo model still retains a remarkable performance to distinguish them.

## Performance of AI-Endo model on validation datasets with different surgeons and skills

The advantages of a learning-based framework are substantially attributed to its ability to recognize surgical actions and learn intrinsic features from surgical video data. For AI-Endo, its modeled spatial embedding and temporal relationships enable it to address various situations. For evaluation on different surgeons, we have tested the AI-Endo model on 15 external patients conducted by three endoscopists with different levels of ESD experience. The model yields an average accuracy of 90.93% (CI: 88.52%, 93.33%) for Surgeon A (6 years experience), 92.93% (CI: 89.81%, 96.04%) for Surgeon B (3 years), and 92.28% (CI: 82.96%, 100.0%) for Surgeon C (2 years). Phase-wise metrics on these 15 cases are provided in Supplementary Table 2. Specific results for each case conducted by these three surgeons are shown in Fig. 3a. These results on different endoscopists demonstrated the generalization capability of the AI-Endo method to accommodate the variation in skill levels of ESD procedures. Such variation affects the proficiency and smoothness of the procedure, which can be reflected by the duration of each surgical phase and the transition frequency between them (see Fig. 3b). In addition, the ESD instruments used in the external validation data are not identical to the expert developmental data, because the design and utility of ESD instruments were evolving over time. As examples illustrated in Fig. 3c, the ESD knives from the developmental dataset included dual knife, insulated-tip (IT), and triangular tip (TT) (Olympus Medical Corporations, Tokyo, Japan), while the external validation dataset also used the updated needle-type knife besides the dual knife and IT. The AI-Endo model can overcome such variation with stable performance regardless of different instruments, showing that its discrimination capability mainly relies on understanding dynamic surgical actions rather than instrument appearances.

For validation on another four cases that involve operation skills that are unseen in the developmental data, AI-Endo shows an average accuracy of 93.07% (CI: 83.44%, 100.0%) on cases with pocket creation method. AI-Endo retains the ability to recognize surgical phases in the pocket creation process, even though the pocket creation is relatively new and not included in our developmental dataset. This advantage is largely attributed to its potential to capture features of tissue background and tissue-tool interactions, which are shared between conventional operations and pocket creation. The accuracy on ESD with line-assisted traction was lower at 75.22% (CI is not calculated for one case). The limitation in accuracy was caused by the emergence of new functional tools (Fig. 3d) during traction application, which coincides with our expectation because it is challenging to be applicable to a specialized tool that looks very different from others. Our model predicts phase Idle for the frames involving this tool while predicting correctly for other frames in general.

## Ex vivo animal study for validation of AI-Endo model

Existing works on surgical phase recognition have not yet clearly investigated the incorporation of the AI-Endo model into clinical workflow, therefore, we designed an ex vivo animal study to optimize and validate the proposed framework in our work, ranging from the layout of third-party monitors to the design of the graphical user interface. Compared to conducting an in vivo animal study directly, adopting a preliminary ex vivo study first is more cost-effective to ensure the AI assistance could deliver useful data analytic results and alleviate interruptions caused by add-on AI functionality. To confirm how to seamlessly integrate the AI-Endo computational tool into the Endoscopy System, we implemented the whole system in a training laboratory at CUHK Jockey Club Minimally Invasive Surgical Skills Centre. Specifically, after the ex vivo porcine colon was cleaned by water lavage, it was fixed within a plastic tray, then an overtube was attached to the colon to simulate the environment inside the colon. Figure 4a shows the entire system pipeline and data flow, i.e., the surgical operation on the animal model was imaged by an endoscope and streamed by an endoscope processor; the video is imported to the

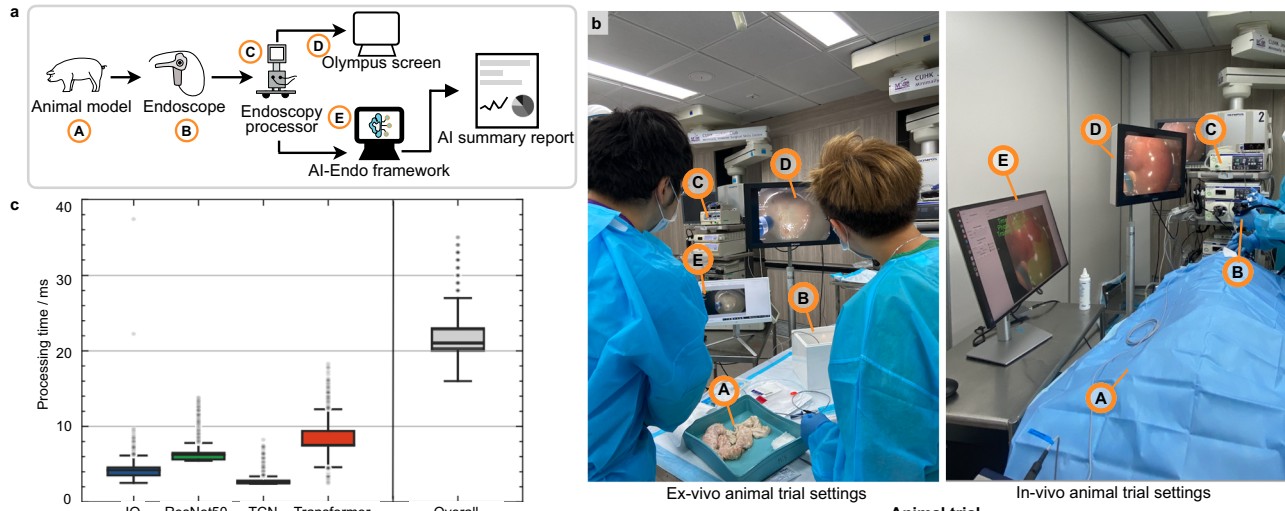

Ex-vivo animal trial settings                                In-vivo animal trial settings

**Animal trial**

**Fig. 4 | Experimental settings and real-time performance of pre-clinical animal experiments. a** The data flow of the entire system which integrates AI-Endo with the existing clinical Olympus Endoscopic system. Each individual component is correspondingly marked in (**b**) for both the ex vivo (left) and in vivo (right) experimental settings, with "A" as animal model, "B" as endoscope, "C" as endoscopy processor for video streaming, "D" as existing Olympus screen, "E" as AI-Endo third-party screen which delivers data analytical results. In addition, **c** shows the processing time for the overall system and the breakdown of each individual key technical part. Results were calculated based on $n = 13{,}341$ frames. The box indicates the median as a line in the box and excludes the upper and lower 25% (quartiles) of data, and the whiskers extend from the box by 1.5 × the inter-quartile range (IQR). Source data are provided as a Source Data file.

AI-Endo model and the automatic analytical results are displayed to surgeons in real-time.

We put a third-party monitor (aside from the existing displaying screen of the endoscopic view) for surgeons to visualize the AI-predicted surgical phase on the screen, in which the surgical phase was overlaid to each frame in top left corner without occluding the main surgical scene (see Fig. 4b). We measured the computation overhead for I/O data flow, which totally took 4 ms for data flow input (i.e., importing video stream from existing surgical system to AI-Endo) and output (i.e., displaying the AI-Endo prediction phase to screen for surgeons to visualize). The AI-Endo model inference took 17 ms, consisting of 6 ms for the ResNet50 module, 3 ms for the Fusion module, and 8 ms for the Transformer module (cf. details of the AI model architecture in "Methods"). Note that the transformer module uses the most time because it needs to aggregate crucial spatial-temporal information for maintaining recognition accuracy. Overall, the efficiency of the entire AI-Endo recognition system reached 47 fps, which can satisfy the requirement for real-time use, without feeling of visual latency. Two stations were set up using the above-described ex vivo setting, with each serving two novices in the training session. For the four trainees, our AI-Endo yielded an average accuracy of 88.88% (CI: 79.95%, 97.82%) over a total of four cases, showing potential to apply the AI model in a streamed ex vivo setting as a holistic system. Phase-wise metrics indicate high sensitivity and specificity regarding the key phases of Injection and Dissection (Supplementary Table 3).

**In vivo animal study on live pigs for validation of AI-Endo model in pre-clinical setting**
Quantities of works have been proposed for automated surgical phase recognition, however, none of them incorporated in vivo animal trials to demonstrate the clinical application of system in real-world surgery. Based on the success of ex vivo animal experiments, we further conducted an ESD surgical training session with in vivo live animal trials, aiming to showcase the clinical applicability of an intelligent phase recognition system with online score analysis and automatic performance report generation. The real-time system integration and data flow of the in vivo experiment were the same as that of the ex vivo experiment.

To support the clinical usage of AI-Endo, we packaged the AI-Endo as a desktop software that seamlessly operates with prevalent surgical settings. The accessibility of AI-Endo becomes much reachable for endoscopists who are more likely to demand a ready-made implementation with a user-friendly graphical interface. In the animal trial, multiple 2-cm-sized lesions were marked for simulated ESD on three different locations in the digestive tract, i.e., rectum, stomach, and esophagus. Twelve ESD procedures were performed on two live pigs, i.e., including five (1/2/2 for rectum/stomach/esophagus) and seven (2/3/2 for rectum/stomach/esophagus) were conducted by an experienced and a novice endoscopist, respectively. The AI-Endo delivered an average accuracy of 83.53% (CI, 81.48–85.58%) over all the in vivo procedures. The relative performance degradation was postulated to be due to anatomical differences between pig and human tissue, as well as the experimental setting of fake lesions. Fortunately, for the Dissection phase which is the most important step for ESD, the AI-Endo achieved a specificity of 91.57% (89.89%, 93.24%) and sensitivity of 86.68% (CI, 83.22%, 90.14%) (Supplementary Table 4). Additionally, AI-Endo achieved accuracy rates of 83.29% (CI: 77.43%, 89.15%), 83.05% (CI: 78.11%, 87.99%) and 84.31% (CI: 78.77%, 89.85%) on rectum, stomach and esophagus, respectively, showing slight differences among different GI organs.

The in vivo animal experiment aimed to serve as a promising pilot study to explore the applicability and capability of AI-Endo for cognitive assistance in real-time complex surgery. In this regard, we tried to derive meaningful skill assessment scores from AI-based workflow recognition results, to automatically analyze the operational skills of novices during the training session. As shown in previous works[44], the surgeon with a higher level of surgical skill tends to operate the surgical tools more smoothly, which benefits from their clear plan on the trajectory of surgical tools and the resection surface. To some extent, the smoothness of the operation can be reflected by the frequency of hesitation and the exchange of surgical tools[45], which can be quantified by the frequency of the surgeon changing across phases. In ESD training session, it is useful to monitor their operational skill in real time, which could reflect their learning curve. In this regard, the AI-Endo system dynamically counted the number of transitions among surgical phases, e.g., the transition from Dissection to Idle when the

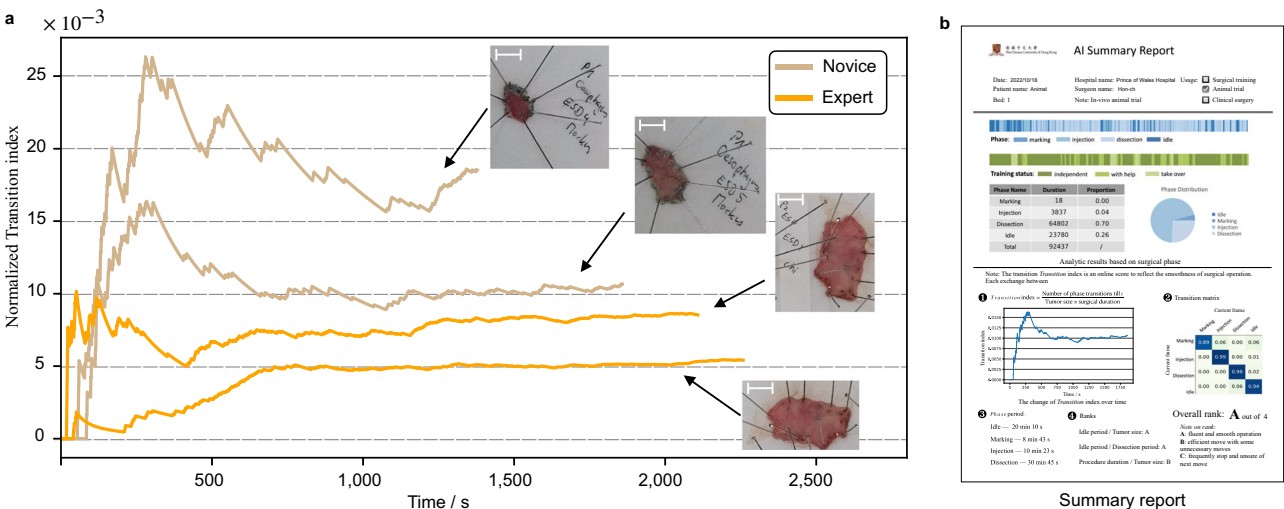

**Fig. 5 | Data analytical results derived from AI-Endo phase recognition for in vivo animal experiments. a** The curves of derived online score of the Normalized Transition index calculated for the senior surgeon (2 cases with orange lines) and the novice (2 cases with brown lines) at the esophagus. Inserted photos represent the dissected samples and the scale bar corresponds to 1 cm; **b** Design of the AI summary report that is automatically generated by the AI-Endo system in preclinical trials. Source data are provided as a Source Data file.

knife retracts. As the size of the lesion could affect the total duration of procedure[46], we divided the total transition frequency by the length of the tumor to remove its bias on the number of phase transitions. The proposed online surgical score Normalized Transition index (NT-index) is defined as the division of transition number over time and the size of the lesion, which yields an NT-index curve to describe the dynamic changes of transition frequency as the operation proceeds. The lower this curve is, the higher the endoscopist's skill level would be. In Fig. 5a, we present the index curves of four in vivo surgical cases, two of which are respectively conducted by the experienced and the novice endoscopist. The analytic results show that the NT-index curves of the senior are generally lower than those of the novice. At the end of the procedures, the senior and the novice get normalized transition index scores on the rectum, stomach and esophagus as (13.94 vs 21.71), (4.39 vs 10.72) and (10.23 vs 16.85). The proposed online score NT-index shows a statistical difference ($p = 0.048$) in the level of ESD skills, which is consistent with our expectations according to the animal trial settings. Based on the index curve, expert endoscopists, e.g., the trainer in surgical training, could provide advice and supervision on specific surgical steps.

In addition, we propose to automatically generate an intelligent report, which summarizes and presents the surgical workflow analytical results to the endoscopists. As shown in Fig. 5b, the summary report intuitively visualized the duration and ratio of each phase. Different from the manual annotation or repeated derivation in previous works[9,36], the AI-Endo instantly provides the endoscopist with an overview of the surgical process and also details the factors that might reflect the surgical skill, such as the duration of phase periods and their corresponding ratios in each endoscopist. The proposed online score of the Normalized Transition index, together with several straightforward offline scores added in the summary report, is supposed to serve as an essential reference for the investigation of procedural knowledge and decision-making skills, taking a leap forward to the potential clinical applicability of AI-Endo.

### Multi-center validation on data from different endoscopic systems and country

To broaden the application of AI-Endo, it is interesting to observe its generalizability to different endoscopic systems and multi-centers. We assessed the performance of AI-Endo using four cases from Nanfang Hospital, Southern Medical University in Guangzhou, China. These cases were conducted using the Fujifilm endoscopic system, which differs from the Olympus endoscopic system utilized in our developmental dataset. To evaluate the potential of AI-Endo in an international cohort, we further tested AI-Endo on an additional four cases from Internal Medicine III-Gastroenterology, University Hospital of Augsburg, Augsburg, Germany. These cases were recorded with the Olympus system but yielded geographical variations across different countries.

We utilized AI-Endo to process the four cases from Nanfang Hospital, Southern Medical University in Guangzhou, China. All cases were annotated and processed in the same manner as the developmental dataset. AI-Endo finally yielded an average accuracy of 90.75% (CI: 88.50%, 93.01%) and exceptional ROC curves for each phase (Fig. 6a). All phase-wise performance metrics were higher than 88% (Fig. 6d). This investigation shows that AI-Endo's performance is robust and generalizable across different endoscopy systems, which aligns with our expectations regarding ESD surgical settings. During endoscopic procedures, conventional white light images are used and these images remain largely consistent across different brands of endoscopes. Additionally, the design and implementation of intelligent algorithms in the development of AI-Endo did not depend on assumptions about the type of instrument being used. AI-Endo can accept the video stream and process data in a relatively independent manner, which means the inference speed should not be heavily dependent on the endoscopy system.

Then, four cases from Internal Medicine III-Gastroenterology, University Hospital of Augsburg, Augsburg, Germany were used to showcase the robustness of AI-Endo under geographical variations. Although the cases were conducted at an international center, AI-Endo maintained its high performance and achieved an average accuracy of 87.34% (CI: 84.43%, 90.25%), specificity of 86.01% (CI: 71.48%, CI: 96.27%), and average sensitivity of 86.60% (CI: 74.21%, 96.36%). AI-Endo delivers promising ROC curves on four phases with AUROC values exceeding 90.67% (Fig. 6b, d). Based on the multi-center dataset from Guangzhou (China) and Augsburg (Germany), we further statistically analyzed the performance of AI-Endo on different organs, including esophagus, colorectum, and stomach, on which AI-Endo keeps a large average accuracy higher than 86.68% (Fig. 6c). These findings suggest that AI-Endo can robustly handle multi-center cases regardless of differences in their geographical or tumor locations, indicating the potential of AI-Endo for wide applications across international medical centers.

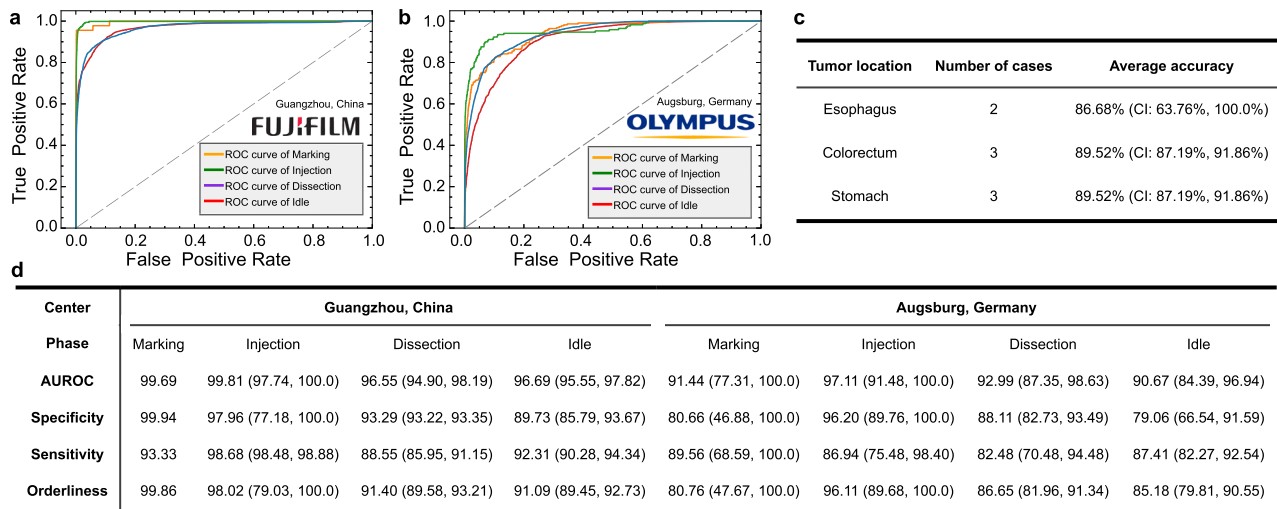

**Fig. 6 | Experimental results on multi-center validation datasets from Guangzhou (China) and Augsburg (Germany). a** The ROC curve of AI-Endo on cases from Guangzhou, China (*n* = 4 cases); **b** The ROC curve of AI-Endo on cases from Augsburg, Germany (*n* = 4 cases); **c** Average accuracy on cases of esophagus, colorectum, and stomach in the multi-center datasets; **d** Phase-wise performance metrics of AI-Endo on multi-center datasets. Data are presented as 95% confidence interval (CI) when applicable. CI is not calculated for the Marking phase in cases from the center in Guangzhou, China, because only one case involves marking. Source data are provided as a Source Data file.

## Discussion

This work aims to investigate intelligent surgical phase recognition from bench to bedside. We established a high-quality ESD dataset of expert operations, together with a well-defined annotation protocol for surgical phase recognition. Based on it, we developed the AI-Endo model to recognize surgical phases with representative spatial-temporal features, achieving high performances on both developmental and external validation datasets. This demonstrates that the AI-Endo model trained on expert data is applicable to junior surgeons with different skill levels, various cases with different ESD techniques and endoscopic systems. More importantly, the AI-Endo was seamlessly integrated into pre-clinical settings, and validated with ex vivo and in vivo animal trials in real-time. The system showed stable performance, and analytical results were delivered to surgeons through a user-friendly interface for intraoperative cognitive assistance and postoperative training assessment.

ESD is a novel endoscopic surgical procedure for complete tumor resection to cure early gastrointestinal (GI) cancer, which is the most common cancer worldwide. Although ESD has good peri-operative outcomes regarding high-rate of en-bloc resection and low rate of local recurrence, the surgery is still challenging with a long learning curve for novices. It is clinically desired to use AI techniques that can learn from expert experiences and data for understanding surgical contexts and further identifying, preventing, and mitigating safety-critical events in operation. To begin with, surgical phase recognition is the fundamental task, i.e., only after the ongoing surgical step is automatically recognized can the smart system conduct subsequent functionalities. Existing works have not systematically investigated this key task due to the lack of expert data, algorithmic limitations, and insufficient pre-clinical validation. This study plays a pioneering role to raise attention and inspire solutions for AI-assisted ESD.

As observed from experimental results, our AI-Endo model successfully addressed the dilemma between accuracy and efficiency for surgical workflow prediction in ESD. Using an inference computer equipped with an Intel Xeon(R) 3.7 GHz CPU with one NVIDIA GeForce RTX 3090 GPU, the model is able to yield a good online deployment accuracy at 47 fps. Noting that such an efficiency includes time spent throughout data analytics in the integrated system, rather than the AI model computation itself. Given that the raw data streaming in the existing Olympus system is maximum at 50 fps, from our human feedback, we did not feel visual latency when using the provided user interface. This demonstrated that the AI model can fulfill real-time requirements given the hardware support of a standard workstation-level configuration. It could suggest the potential of applying advanced surgical AI tools in low-income countries.

Regarding how to properly incorporate the AI-Endo model into the existing clinical workflow, we in fact had multiple rounds of discussion and optimization among engineering and clinical team members. Existing literature on computer-assisted surgery in general has not yet clearly investigated this important issue. Basically, we think that at least two points should be considered for the integrated system design. The first is to ensure that the system delivers useful data analytic results that are otherwise not obtainable without AI assistance. The second is to avoid the add-on AI functionality changing the surgeon's operation habit in the current routine. In these regards, we propose to display the AI predictions on a third-party screen putting it side-by-side to the existing Olympus screen. The ongoing surgical phase is monitored by the AI-Endo behind the curtain, which presents steady progress of the procedure. More importantly, we derive an online score based on the surgical phase recognition for skill assessment and apply it to the ESD training session. This score is automatically calculated to reflect the proficiency and smoothness of ESD. Despite it is not yet thoroughly validated from clinical usage, we regard this as an inspiring initial step for driving AI's role in facilitating novice surgeons. In our future work, we target integrating the AI-Endo into the endoscopy system as off-the-shelf software, displaying the analytic results on the embedded monitor in a straightforward manner.

Limitations of our work lie in two aspects. First is the relatively small number of cases in developmental dataset, which is actually a common drawback of most existing works on surgical AI. The current largest public dataset, i.e., Cholec80 on laparoscopic cholecystectomy[47], has 80 full-length surgical videos at a high frame rate. The small-scale training data is still not comparable to the big data as used in other deep learning applications such as face recognition and autonomous driving. Fortunately, our collected data was of high quality in terms of expert skill level, long-time expansion, various dissection locations, and diverse surgical scenes, which helped to compensate for the shortage. The clearly defined

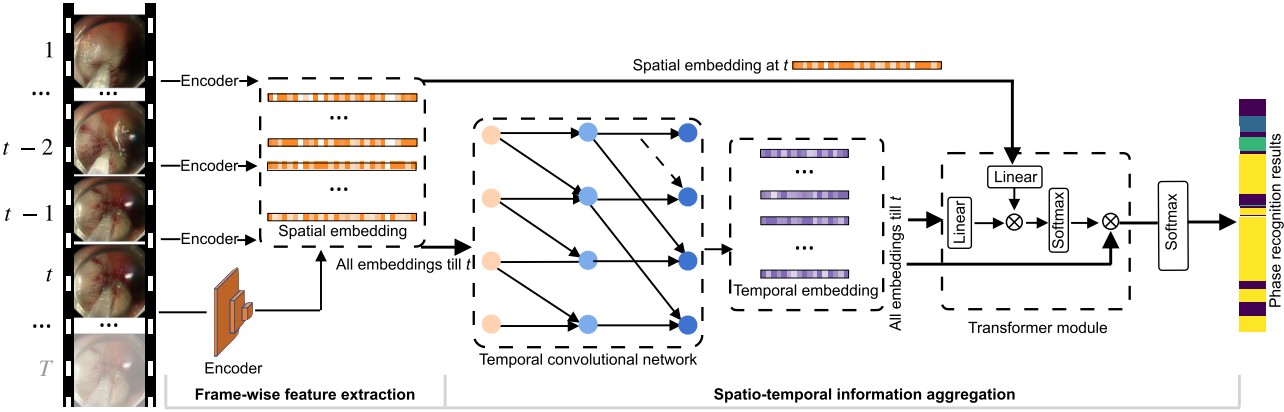

**Fig. 7 | The architecture of AI-Endo deep learning model for real-time recognition of surgical phases.** Each frame of video stream is sequentially encoded by ResNet50, followed by a temporal convolution network to fuse spatial-temporal information. Thereafter, spatial embedding at $t$ is used as the query in the transformer-based module for predicting the frame-wise surgical phase. Different colors represent different feature embeddings or output values.

annotation protocol was also important to ensure the labeled 0.2 million training frames were consistent as ground truths for model learning. The second limitation of this work concerns the model generalizability, which was noticeable from the performance drop observed in ex/in vivo animal experiments (Supplementary Tables 3 and 4). Despite this being explained by the appearance difference between animal tissue and human tissue, similar degradation is anticipated to be encountered under the emergence of new tools (Fig. 3d) that were not covered by the developmental data. A relatively small dataset limits the model's robustness to identify effective tool features or surgical scenes when an unseen ESD technique is involved. Our currently developed method has not particularly addressed this problem, while can be extendable with domain generalization[48] and test-time adaptation[49] strategies. Promisingly, the proposed model has shown a noteworthy degree of adaptability to the variations encountered in surgical settings, such as differences in geographical locations and endoscopy systems. This matters in its wider application and multi-center deployment in the future.

Last but not least, future works of this study will continue to focus on AI assistance for ESD. The benefits of automatic phase recognition go beyond the generation of statistical reports and the calculation of online NT-index, which provide only a limited view of surgical skill evaluation. We encourage community researchers to utilize the open-source code and data we provide to explore the statistical significance of surgical phases and promote progress in surgical training and related areas, such as establishing large-scale structured and segmented surgical phase databases[50]. Besides, based on the high-performance surgical phase recognition, we will extend the video analysis to semantic segmentation of surgical scenes such as the submucosal layer, muscle layer, and vessels in our future work. We implemented a preliminary segmentation model in this study's in vivo animal experiment, which also fulfilled real-time prediction speed. We will further improve its accuracy and accordingly investigate how to use it to help surgeons reduce adverse events on safety-critical tissues. Moreover, AI-enabled data analytics would provide cognitive assistance and support decision-making for surgery, which has a large potential to enhance surgical safety. As artificial intelligence is increasingly investigated for surgical applications, its way of integration in the operating room and clinical role for benefiting surgeons are to be emphasized along the way. We aim to include clinical trials in our future work after the entire system is extensively validated with more surgeons and clinical centers, ensuring participant safety in invasive procedures.

## Methods

### Data collection

In this study, developmental dataset was collected from Prince of Wales Hospital in Hong Kong and validation datasets were gathered from Prince of Wales Hospital; Nanfang Hospital, Southern Medical University in Guangzhou, China; and Internal Medicine III-Gastroenterology, University Hospital of Augsburg in Augsburg, Germany. All patient information in these retrospective cohorts was de-identified, and only the imaging system and surgeon's name were kept for data analysis. Ex vivo and in vivo animal cases were conducted during the animal trial sessions at CUHK Jockey Club Minimally Invasive Surgical Skills Centre. Ethical approvals were obtained from the Ethics Committee of The Chinese University of Hong Kong (No. 22-145-MIS).

### Problem formulation and network learning

Given an ESD video stream, this work formulates the phase recognition task as an online classification task based on our previous work[20]. Given a video stream $V = \{x_t \in \mathbb{R}^{H \times W \times 3}\}_{t=1}^{T}$ with $T$ frames, we make the phase recognition model as a function $\mathcal{F}_\theta$ which classifies each frame $x_t$ into one of four surgical phases according to probability prediction $p_t = \mathcal{F}_\theta(x_1, x_2, \ldots, x_t)$, where each element represents the probability of frame $x_t$ being phase in {Marking, Injection, Dissection, Idle}. Due to the complexity of recognizing surgical phases with large intra-class variation and inter-class similarity, we decompose $\mathcal{F}_\theta$ into two stages $\mathcal{F}_\theta = \mathcal{G}_\omega \circ \mathcal{H}_\phi$, with $\mathcal{G}_\omega$ as the feature extractor to encode discriminative representation for each single frame, and $\mathcal{H}_\phi$ as the follow-up spatial-temporal feature aggregator for yielding the final phase prediction incorporating video dynamics. An overview of our AI-Endo network is illustrated in Fig. 7.

In ESD surgery, the differences in anatomical structures and lesion locations introduce considerable intra-class variances on $x_t$, imposing challenges on $\mathcal{G}_\omega$ to learn discriminative frame-wise representations, which are the basis for spatial-temporal feature learning. We propose to rely on self-supervised learning with contrastive loss in the training process, by formulating $\mathcal{L}_{con}$ (see Eq.1) which enhances the similarity of embeddings from intra-class frames (a.k.a., positive pairs) while enlarging the distance between inter-class frames (a.k.a., negative pairs). The embedding $e_i$ for each frame $x_i$ is extracted using a pretrained ResNet50[51] as backbone. Meanwhile, in order to enhance the discrimination capability of learned features toward the phase recognition task, we also add cross-entropy loss with respect to phase labels annotated for each frame. In these regards, the overall loss function training $\mathcal{G}_\omega$ is as

follows:

$$\omega^* = \arg\min_{\omega} \mathcal{L}(\mathcal{G}_{\omega}; \{x_i, y_i\}_{i \in I}) = \mathcal{L}_{con} + \mathcal{L}_{ce},$$

$$\mathcal{L}_{con} = \sum_{i \in I} \frac{-1}{|A(i)|} \sum_{n \in A(i)} \log \frac{\exp(e_i \cdot e_n / \tau)}{\sum_{a \in N(i)} \exp(e_i \cdot e_a / \tau)}, \quad (1)$$

$$\mathcal{L}_{ce} = \sum_{i \in I} \sum_{k=1}^{4} \mathbb{1}\{y_i == k\} NLL(\mathcal{G}_{\omega}(x_i), y_i),$$

where $i$ denotes the index of frames in the mini-batch $I$. $A(i)$ and $N(i)$ respectively represent the frames that have the same and different phase annotations with $x_i$, and $\tau \in \mathbb{R}^+$ denotes the scalar temperature parameter[52]. The $\mathbb{1}\{y_i == k\}$ is the label indicator for the negative log-likelihood function and equals 1 when $y_i = k$ otherwise 0. In addition, to facilitate real-time deployment, the pretrained feature backbone was pruned by removing the two linear projection heads. The contrastive learning strategy enables the remained modules to still provide meaningful embedding without increasing the computation overhead. The final embedding for each frame was sequentially used as the input for the subsequent spatial-temporal feature learning.

Temporal reasoning is essential for AI-Endo to capture dynamic information in the procedure, such as the trajectory of surgical tool and its interaction with the targeting tissues. In this regard, we leverage a fusion module to extract long-range temporal information with a temporal convolution network (TCN). In order to aggregate the spatial and temporal information and boost the capability of representation, we further incorporate a global attention-based transformer module to capture supportive relationship based on spatial and temporal embeddings.

For the fusion module, we use TCN to perform hierarchical refinement on temporal embedding. Given the spatial embedding sequence $\{e_i = \mathcal{G}_{\omega}(x_i) \in \mathbb{R}^d\}_{i=1}^t$, the TCN targets generating temporal embedding by exploring inter-frame relationship. The TCN is composed of multi-level temporal convolution layers, each level of which includes consecutive dilated residual layers. Taking the $l^{th}$ layer as an example, the output $D_{l+1}$ is calculated by $D_{l+1} = D_l + W_{2,l} * \{ReLU(W_{1,l} * D_{l-1} + b_{1,l})\} + b_{2,l}$, where $W_{1,l}$ and $W_{2,l}$ are the weights of dilated convolution and $1 \times 1$ convolution, whose biases are denoted as $b_{1,l}$, $b_{2,l}$, respectively. The first layer accepts $D_0 = \{e_i\}_{i=1}^t$ as the initial input. To get a larger inception field of temporal convolution, we gradually increase the dilation factor by 2, which yields an increased size of the inception field. The output $D_l$ is shifted along the temporal dimension so that the output $D_{L+1}$, i.e., the temporal embedding $m_i \in \mathbb{R}^{d'}$, only relies on current and previous frames.

Although the fused spatial-temporal embedding at $t$ integrates the temporal information at neighboring frames, a fixed-size embedding representation is insufficient to deliver complicated information in both dimensions of time and space. Therefore, we rely on the transformer module to obtain the phase prediction by further aggregating spatial and temporal information with global attention. Specifically, we take the spatial embedding $e_t$ as the query and the temporal embeddings $M_t$, i.e., concatenation of $\{m_i\}_{i=t-n+1}^t$, as the key and value, where $n$ denotes the range of selected temporal embeddings before time point $t$. The spatial embedding $e_t$ is first reduced to $\hat{e}_t$ with the same dimension as that of $m_i$ through linear projection. Then $\hat{e}_t$ and $M_t$ are processed by transformer as:

$$Trans(\hat{e}_t, M_t) = softmax\left(\frac{W_q \hat{e}_t \times (W_k M_t)^T}{\sqrt{d'}}\right) W_v M_t, \quad (2)$$

where $W$ are the linear projection mapping metrics and $p_t = softmax(Trans(\hat{e}_t, M_t))$ yields the final phase prediction. The fusion module and transformer module can be trained end-to-end to derive the optimized model $\mathcal{H}_{\phi}^*$. The trained model is capable of extracting long-range spatial-temporal information.

## High-throughput online prediction

The application of this framework requires efficient deployment coupled with intraoperative video streaming. To achieve this goal, we reduce the computation complexity by analyzing how the feature embeddings at each frame are updated according to the inception field of the AI model. For the fusion module, rather than continuously storing all the spatial embeddings $\{e_i\}_{i=1}^t$ for the temporal reasoning of TCN, we only selectively keep the embeddings within its inception field. Concretely, given the inception field of TCN is 512, the spatial embedding $e_{t+1}$ only interacts with 511 previous frames, i.e., accounting for over 10 s under our inference speed of 47 fps. We build a first-in-first-out (FIFO) queue to dynamically store $\{e_i\}_{i=t-510}^{t+1}$. When the spatial embedding $e_i$ is out of the inception field, it graduates from the queue. Notably, this framework keeps high inference efficiency and also fully preserves its accuracy.

For the model training, all cases of the developmental dataset were first arranged in chronological order, and then we sampled them at five equal intervals. This procedure resulted in 5 folds, with four folds used for training and the remaining one for testing in a cross-validation manner. The framework was optimized in two separate stages, i.e., training feature embedding $\mathcal{G}_{\omega}$ and spatial-temporal information aggregation $\mathcal{H}_{\phi}$. At the first stage, $\mathcal{G}_{\omega}$ was trained for 8000 iterations with batch size 128. The learning rate started from $5e^{-4}$ and was reduced by 10 after 6000 iterations. When the first stage of training was finished, we fixed and utilized the trained model $\mathcal{G}_{\omega}^*$ to generate the spatial embeddings of all frames for training $\mathcal{H}_{\phi}$. At the second stage, the model $\mathcal{H}_{\phi}$ was trained for 4000 iterations by selecting all consecutive frames in a video as the input in each iteration. By adopting temporal convolution[53], the model was empowered to process all feature embeddings of the video in a causal manner, thus preserving the characteristics necessary for online prediction. We set the learning rate as $5e^{-3}$ at the beginning and multiplied it with 0.1 at iterations 1500 and 2500. The parameters of modules $\mathcal{G}_{\omega}$ and $\mathcal{H}_{\phi}$ were both optimized by SGD with momentum. Supplementary Fig. 5 shows the curves of training loss, where the loss curves became flat at the end of the training process. Therefore, models at the final iteration were used for phase prediction. After optimizing and fixing the network structure and hyper-parameters, we proceeded to retrain the model using the entire developmental dataset. This was done to maximize the amount of available training data and improve the model's generalization performance[54]. For any future applications of AI-Endo on other datasets, such as external, ex vivo, and in vivo animal studies, we used the model that was trained on the entire developmental dataset.

## Description of the animal studies

Study design of live animal experiments. Two healthy female pigs of ~30kg were used as the in vivo porcine models for ESD experiments under general anesthesia at CUHK Jockey Club Minimally Invasive Surgical Skills Centre (CUHK MISSC). The procedures were performed with a high-definition endoscope (GIF-H190 with straight transparent hood, Olympus Medical Corporation, Tokyo, Japan) and ESD knife (Dual knife J, Olympus Medical Corporation, Tokyo, Japan). The VIO3 (Erbe Elektromedizin GmbH, Germany) was used as the electrosurgical power platform. During the ESD procedures, circular lesions were pre-marked in the porcine esophagus, stomach and rectum with 2 cm in diameter for subsequent ESD simulation in aminal experiments. Due to the increasing difficulty in performing ESD in the stomach, esophagus and rectum, the time required for each procedure also increased accordingly, especially for novice endoscopist. As a result, the number of procedure performed by each endoscopist were different. Specifically, the experienced endoscopist performed seven procedures, including 3 stomach, 2 esophagus and 2 rectum, while the novice endoscopist performed five procedures, 2 stomach, 2 esophagus and 1 rectum. Such a design of animal experiments can cover diverse scenarios, therefore allowing us to observe the AI model's efficacy in

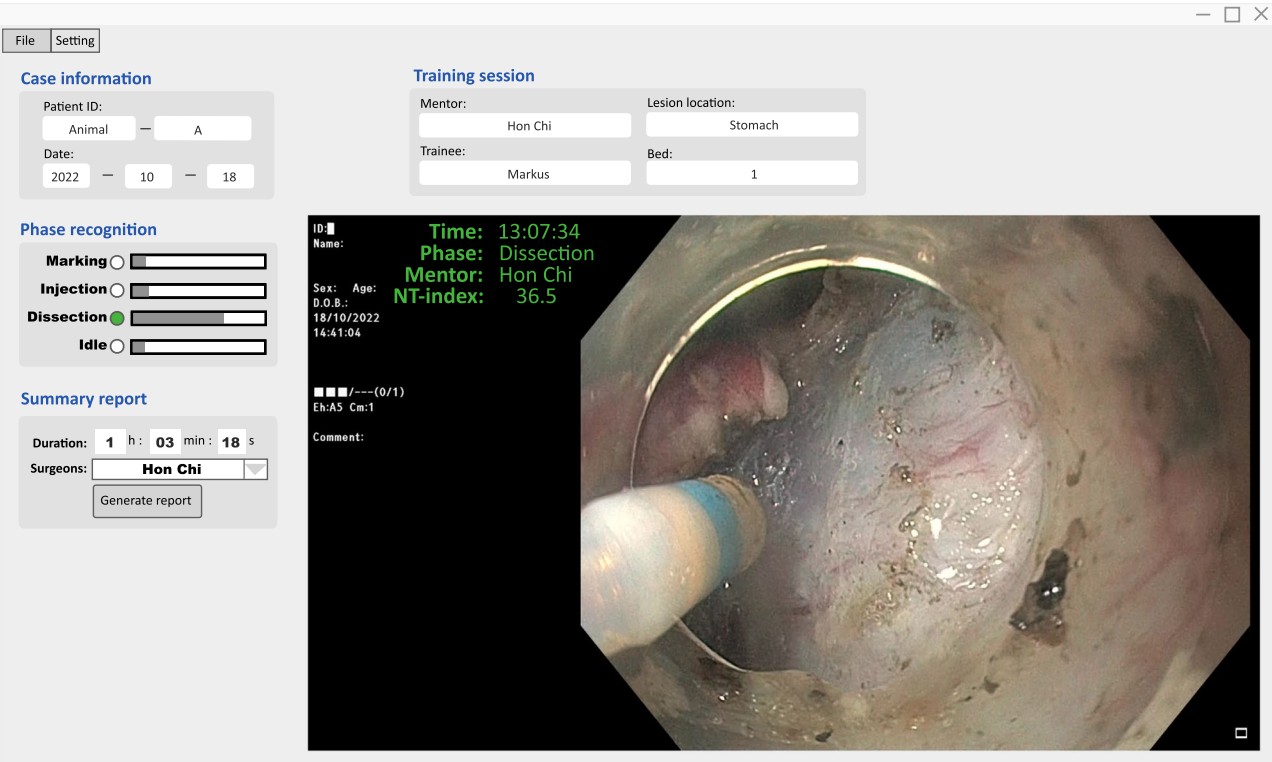

**Fig. 8 | The desktop software of AI-Endo.** The user interface includes basic information, phase prediction, AI result display and summary report generation button. The software is integrated into real-time clinical settings.

general. Consent was obtained from the endoscopists to publish identifiable information as shown in Fig. 4b.

To integrate AI-Endo into real-time surgical workflow, we packaged the algorithms as a ready-to-use software providing automatic data analytics and an interactive user interface (illustrated in Fig. 8). It was deployed on a standardized workstation with an Intel Xeon(R) 3.7 GHz CPU and one GPU of NVIDIA GeForce RTX 3090. The video stream from the Olympus endoscopy system was exported from the SDI port and converted through the SDI to a USB converter (U3SDH, ACASIS, China), then imported to the AI-Endo software. This data was used as input to the network for intelligent workflow recognition, and the predicted outputs were visualized through the user interface. Our AI-Endo adopted a third-party monitor thus not changing the operation style of an existing clinical setting. The user interface was mainly composed of three parts, including procedure basic information, phase recognition result and intelligent skill summary. Specifically, using the AI-Endo software in practice, we can mark clinical basic information, e.g., patient ID, surgeon name, lesion location and size, operation date, etc. As the procedure starts, the AI-Endo would automatically recognize the ongoing surgical phase at each time point, and overlay the results onto video frames dynamically. The computation speed can achieve 47 fps using the standard workstation, which sufficiently satisfies the requirements for real-time application. In live experiments, the AI software could monitor the progress of the ESD procedure, and timely reflect the smoothness of dissection, which was useful for the mentor to easily track the practicing status of trainees. We noticed that the AI-Endo was sensitive to the actions of surgical instruments and their interaction with target tissues, which was reflected in online predictions showing frequent transitions between dissection and idle. Upon finishing the entire procedure, our AI-Endo software can automatically generate a structural report to statistically summarize the surgical workflow, and give an objective assessment of the training session. All the developed functions are easy to use for surgeons without the need for coding experience. Both the mentor

and trainee were satisfied with the AI-Endo software design and its way of incorporation into the existing operation system.

The AI-Endo could generate an intelligent report that presents statistical information and a structural summary of training performance (Fig. 5b). Taking advantage of automatic data analysis, the performance of the surgeon could be assessed immediately in an objective way. The specific content of the report includes basic information, phase statistics and skill assessment. First, the basic information shows the date, hospital, case name, endoscopist name, training session and settings. Second, the phase statistics section visualizes the entire procedure using a color bar, where the frames of each phase are marked in different colors over time, based on automatic recognition results. Meanwhile, the training status was shown aside, indicating the degree of guidance that the trainee received from the mentor, i.e., independent, with help, or take over. As a quantitative analysis, we calculated the duration distribution across four phases, which is useful for the overall understanding of the surgical skills for ESD[9,36]. We got the percentage of each phase by counting the number of corresponding frames from AI predictions, with the calculated ratio visualized in an intuitive pie chart. Third, our AI-Endo software further analyzed the skill-aspect performance of endoscopists based on phase recognition results. Online score of NT-index and offline scores of transition metrics and phase periods (which can be straightforwardly derived from the above statistics) were reported. The curve of NT-index was plotted along time in an overview format. The transition matrix was added to show the inter-changing frequency among phases, which reflects the proficiency and smoothness of the operator. Phase periods listed time duration of each phase, which was further used for calculating the Idle period/Tumor size, Idle period/Dissection period, and Procedure duration/Tumor size. Based on these skill assessments, we could compare and rank the performance of endoscopists for the training session. Our experiment demonstrated that this relative comparison matched the mentor's subjective impression of the skill levels of different trainees. The design of the report

template was a result of insights and rounds of discussions from both engineers and clinicians.

## Statistical analysis

All statistical analyses were performed with Python (v3.6). For the quantitative results of the performance on the development and external datasets, we adapted Student's $t$-distribution with 95% confidence interval (CI: lower%, upper%). To compare the analytical results from different groups, we used a two-sided pairwise T-test to inspect their statistical difference. A $P$-value of <0.05 was considered as statistically significant.

## Reporting summary

Further information on research design is available in the Nature Portfolio Reporting Summary linked to this article.

## Data availability

All data supporting the in vivo and ex vivo animal trial studies are publicly available in the Figshare database https://doi.org/10.6084/m9.figshare.23506866.v5. Due to ethical regulations on confidentiality and privacy, access to the human cases used for training and validating models is limited to authorized researchers approved by the ethics committee. The timeframe for the ethics application would be about two months. These data are available from the corresponding authors upon request with justification of specific usage of the data and non-commercial purposes. Source data are provided with this paper.

## Code availability

AI-Endo was implemented with Python 3.6.13 and PyTorch 1.10.2. The source code for this project is available at the GitHub repository https://github.com/med-air/AI-Endo[55].

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

## Acknowledgements

We thank Dr. Alanna Ebigbo, MD, Dr. Andreas Probst, MD, and Dr. Helmut Messmann, MD for help in collecting external validation data from Internal Medicine III-Gastroenterology, University Hospital of Augsburg, Augsburg, Germany. We also thank Dr. Side Liu, MD, for helping on collecting external validation data from Nanfang Hospital, Southern Medical University in Guangzhou, China. This research was partially supported by the Centre for Perceptual and Interactive Intelligence (CPII) Ltd and Multi-scale Medical Robotics Center (MRC) Ltd under the HKSARG's Innovation and Technology Commission (ITC)'s InnoHK Scheme, Hong Kong Innovation and Technology Commission Project No. ITS/237/21FP, Hong Kong Research Grants Council Project No. T45-401/22-N, Guangdong Science and Technology Plan Project No. 2022A1515011477. The funder had no role in the study's conduct, design, data collection, interpretation, or writing the report.

## Author contributions

Q.D., M.L.M., W.Y.C., Y.Y. and H.C.Y. conceived and designed the study; J.F.C., H.C.Y., Y.Y.C., M.S., X.B.L., M.K.C., and W.Y.C. were responsible for collecting and organizing data as well as conducting manual annotations; J.F.C., Y.Y.C., H.Z.Y., Y.H.L., Y.M.J., Y.Y., M.L.M., and Q.D. conducted artificial intelligence framework formulation, deep learning algorithm implementations, experimental design and data analysis; H.C.Y., M.S., and W.Y.C. provided feedback on the developed models from a clinical perspective and conducted animal trials; J.F.C., H.C.Y., Y.M.J., M.L.M., W.Y.C., Y.Y., and Q.D. co-wrote the manuscript with all the other authors providing constructive feedback for revising the manuscript. All authors read and approved the final manuscript.

## Competing interests

The authors declare no competing interests.
