## [Peer Review File · Nature Communications]

Intelligent Surgical Workflow Recognition for Endoscopic Submucosal Dissection with Real-time Animal StudyREVIEWER COMMENTS

Reviewer #1 (Remarks to the Author):

The paper presents a vision-based system (AI-Endo) for surgical workflow recognition in endoscopic submucosal dissection (ESD). It is based on deep learning methods and was trained on an extensive dataset consisting of ESD videos from an expert. It was evaluated on an external dataset as well as ex-vivo animal experiments and in-vivo live animal trials. The system enables a live visualization of on-line workflow recognition during ESD as well as an automatic report regarding the intervention. The application towards endoscopic skill assessment is also shown in the paper.

Relevance

It addresses a relevant topic in the field of surgical workflow recognition which is a basic requirement for multiple assistance functions during such interventions such as context-aware guidance, performance assessment or automatic reporting.

Originality & Soundness

The paper presents interesting and solid work. While the methods that are applied are not completely new, the application is innovative. I especially appreciate the evaluation part consisting of different ex-vivo and live in-vivo trials that are necessary for translation of such systems into clinical practice, but in some aspects it is a bit preliminary for such a venue and there need to be some points clarified. Language, figures and references are sound.

Clarity of presentation

The paper is clearly written and well structured. It provides a detailed introduction to the basic principles of the approach.

I think it is a nice idea and involves a lot of work, but I see some substantial points with it that would argue against a high-impact Nature Communications publication in its current form (see detailed comments).

Detailed comments:

- The training dataset was acquired from only one expert; does the dataset cover all the variety and exceptions that can occur? In addition, it was acquired during a time period of 15 years, I would expect that also the level of expertise is quite different if you compare recordings from the beginning and the end?
- Annotation protocol: how did you define phase transitions, e.g. the start and end frame exactly?
- How were annotation disagreements during the verifications process amongst the ESD surgeons resolved? How often did this occur, e.g. what was the inter-rater agreement?
- While the authors go in great detail in regards to the different metrics for the different phases for the data of the expert surgeon, these details are missing in the evaluation of the external dataset, the ex-vivo and the in-vivo porcine study. To allow for a thorough evaluation and comparison of the performance in the different domains, a table similar to 2b) (or its contents) should be provided for each domain. As the different classes are highly imbalanced, overall accuracy does not give a complete picture.
- Was the same annotation protocol followed for the external dataset, the ex-vivo and the in-vivo porcine study as was for the expert dataset? This is not entirely clear in the paper.
- On what data were the models for the external dataset, the ex-vivo and the in-vivo porcine study trained? Was the entire expert dataset used or was one of the 5 models used for the expert cross-validation-study used?
- How were the final models selected? Did you use the model that resulted after the 4000 iterations of the 2nd stage?
- Could you elaborate further on your "orderliness" metric? To me it seems identical to accuracy (assuming TF is supposed to be TP) and I don't understand how it captures the order of the frames

per phase

- Could you comment on the difference in the percentage of the duration spent in "idle" in expert vs. external dataset?
- In section "Statistical analysis" you state that a significance analysis was performed though I cannot seem to find it in the text
- What do the photos in 5a) show?
- Do you plan to make the datasets publicly available? I would expect this for a high-impact publication as contribution and it would be highly appreciated by the community

Reviewer #2 (Remarks to the Author):

The manuscript entitled "Intelligent Surgical Workflow Recognition for Endoscopic Submucosal Dissection with Real-time Animal Study", you tried to elucidate promising performance of AI-Endo, an intelligent surgical workflow recognition suit, for AI assistance in endoscopic submucosal dissection (ESD).

1. Development dataset and external validation dataset were obtained by human cases and validation of AI Endo was only made by ex-vivo and in-vivo animal models. Further study by using human cases is warranted to show the real performance of AI-Endo.
2. Primary endpoints for a series of pre-clinical studies are unclear. What is the possible intervention by using AI-Endo in clinical use? Did AI-Endo evaluate only the performance of endoscopists?
3. You divided a sequence of ESD into 4 phases, marking, injection, dissection, and Idle. However, marking is only made in the very initial phase, and the others are repeated during ESD. Usefulness of the classification would be confirmed before making annotation and training AI.
4. Injection phase may include not only injection with injection needle. But water injection with electrosurgical knives or the scope tip would be also injection phase.
5. Dissection phase may be divided into mucosal incision phase and submucosal dissection in a narrow sense phase. In addition, how to classify images of bleeding or hemostasis is unclear. Better endoscopists can dissect less bleeding (red color images) and less use of hemostatic forceps (treatment of nonbleeding visible vessels by electrosurgical knives). In addition, better endoscopists would make less exchanges of devices and less proportion of idling time.
6. There are several novel techniques of ESD such as pocket creation method or traction method. Is AI-Endo applicable for these different techniques?
7. Is AI-Endo only applicable for Olympus systems? How about Fujifilm and Pentax systems?
8. The performance would be changeable according to the GI organs (esophagus, stomach, duodenum, colon and rectum). Which organ is the main application of AI-Endo?

Major revision on manuscript #NCOMMS-23-02595

Intelligent Surgical Workflow Recognition for Endoscopic Submucosal Dissection with Real-time Animal Study

The authors would like to thank all the reviewers for taking time to handle our paper and giving positive feedback on our work. We are happy to see that all people find our work interesting, and we agree that the reviewers' suggestions are helpful to further strengthen our work towards an impactful contribution to the community. As a response to reviewers' comments, we have answered all comments point-by-point in this response file, correspondingly highlighting significant changes in the revised manuscript. We highly recommend the reviewers check the manuscript for more details.

Response to reviewer 1

General Comment — “The paper presents a vision-based system (AI-Endo) for surgical workflow recognition in endoscopic submucosal dissection (ESD). It is based on deep learning methods and was trained on an extensive dataset consisting of ESD videos from an expert. It was evaluated on an external dataset as well as ex-vivo animal experiments and in-vivo live animal trials. The system enables a live visualization of online workflow recognition during ESD as well as an automatic report regarding the intervention. The application towards endoscopic skill assessment is also shown in the paper.”

Reply: Thank you for your affirmative comments. We highly appreciate your time in carefully reading our paper in detail and your efforts in providing a clear and concise summary of our work. As you recognized, our work focuses on a vision-based system for intelligent surgical workflow recognition in ESD, which is a popular endoscopic surgery for GI cancer. Different from previous works, we not only studied deep learning methods on an expert dataset, but also did extensive experiments with the integrated AI-Endo system including ex-vivo and in-vivo animal trials. We have validated its clinical usefulness in an ESD training session with real-time visualization of workflow prediction together with an automatic report for statistical analysis. These are important contributions to the field, and with this work, we hope to raise attention for the community to think about how to integrate AI into clinical workflow in the area of AI for surgery.

“Relevance

It addresses a relevant topic in the field of surgical workflow recognition which is a basic requirement for multiple assistance functions during such interventions such as context-aware guidance, performance assessment or automatic reporting.”

Reply: We thank the reviewer for depicting the relevance of the research topic of this work. We fully agree with your point. Surgical workflow recognition is indeed a fundamental research topic, and it serves as a basic requirement for much higher-level cognitive assistance with AI during surgical interventions. Without accurate and real-time recognition of surgical workflow, those more complex functions in context-awareness, surgical skill, and performance assessment or automatic reporting would be challenging to achieve.

“Originality & Soundness

The paper presents interesting and solid work. While the methods that are applied are not completely new, the application is innovative. I especially appreciate the evaluation part consisting of different ex-vivo and live in-vivo trials that are necessary for translation of such systems into clinical practice, but in some aspects, it is a bit preliminary for such a venue and there need to be some points clarified. Language, figures, and references are sound.

Clarity of presentation

The paper is clearly written and well structured. It provides a detailed introduction to the basic principles of the approach. I think it is a nice idea and involves a lot of work, but I see some substantial points with it that would argue against a high-impact Nature Communications publication in its current form (see detailed comments).”

Reply: Thank you for your affirmative comments on our interesting and solid work which is well presented in this paper. It is truly encouraging for us to see reviewers appreciate our huge efforts in carefully and systematically conducting ex-vivo and in-vivo trials for the AI system. We also strongly agree that exploring how to translate current AI techniques into clinical practice is crucial, but unfortunately still not much work investigating this point in the existing literature. We are kind of proposing the first attempt to conduct systematic experiments with ex-vivo and in-vivo trials, by integrating AI-Endo into the Olympus system in a mock-up operating room, based on the use case of ESD procedure. Completing these works was not easy, because it not only involves much practical engineering workload, but also requires close collaboration between engineering and clinical partners. During this work, all our co-authors including experts in AI and surgery have conducted frequent meetings to discuss data annotations and experimental design, as well as did thought-through preparations before the actual trials, including dividing the two steps of ex-vivo and in-vivo components.

However, we admit that our current work is still not perfect, which is also due to the innovative nature of this work and lack of related work for reference, therefore some aspects still need to be further improved. Fortunately, reviewers have provided very good suggestions with constructive comments to help us strengthen our work. In the revision, we have addressed all the suggestions, including quantitatively explaining the variety and high-level performance of expert dataset (Supplementary Note 1, Table 1), reporting complete evaluation results on external,

ex-vivo and in-vivo datasets (Tables 2, 3, and 4), elaborating the training and evaluation details (section *Problem formulation and network learning*), releasing the ex-vivo and in-vivo datasets (section *Data availability*), etc. Please refer to clarifications in the rest of this response file, and details in the updated manuscript which we believe show improved quality as a high-impact paper.

Reviewer Comment 1.1 — “The training dataset was acquired from only one expert; does the dataset cover all the variety and exceptions that can occur? In addition, it was acquired during a time period of 12 years, I would expect that also the level of expertise is quite different if you compare recordings from the beginning and the end?”

Reply: We thank the reviewer for the insightful comments and concerns regarding the training dataset used in our study. Our keeping data from one expert is beneficial for surgical skill consistency for the dataset, which is important for providing a clean high-quality data to train the AI model. To curate the dataset, we worked closely in collaboration with Prof. Philip Wai-Yan Chiu, a world-class expert endoscopist, who provided examples of exceptional surgical performance. Despite being based on the experience of a single expert, the dataset showcases substantial diversity owing to the comprehensiveness and complexity of the procedures involved. We conducted a statistical analysis to assess the diversity of our dataset, which included a breakdown of exceptions (25.53%), lesion locations (6.38%/87.23%/6.38% of rectum/stomach/esophagus), and dissection tools used (72.34%/10.64%/17.02% of dual/isolation-tipped/triangle-tipped knife). Additional details are provided in Table 1 for easy reference and have been included in the supplementary material.

Table 1: Statistical Analysis on the Diversity of Developmental Dataset. (Supplementary Table 1)

Group	Variety	Percentage
Exception	Bleeding	25.53%
Location	Rectum	6.38%
	Stomach	87.23%
	Esophagus	6.38%
Date	2008~2012	25.53%
	2012~2016	38.30%
	2016~2020	36.17%
Dissection tool	Dual knife	72.34%
	Isolation-tipped knife	10.64%
	Triangle-tipped knife	17.02%

Regarding the time period of 12 years (from 2008 to 2020) during which the dataset was collected, we acknowledge the possibility of variations in the level of expertise between the recordings

from the beginning and the end. However, we would like to assure that Prof. Philip Wai-Yan Chiu has consistently maintained a high level of expertise. He conducted the first ESD surgery in Hong Kong and had already completed more than 100 cases on each of organs rectum, stomach, and esophagus by 2008, and based on the learning curve reported in [1–3] (80/30/30 cases of rectum/stomach/esophagus, respectively), Prof. Chiu can be considered as an expert even at the starting point of data collection. Additionally, we conducted a statistical analysis of dissection speed across three time periods, 2008~2012, 2012~2016, and 2016~2020, yielding values of 0.062 cm/min (95% CI: 0.039, 0.084), 0.061 cm/min (95% CI: 0.048, 0.073), and 0.074 cm/min (95% CI: 0.058, 0.089). The t-test results indicate no significant difference among these three stages, demonstrating that Prof. Philip Wai-Yan Chiu has consistently maintained a high level of expertise.

Our aim was to train a model that can effectively handle a wide range of surgical situations and contribute to the advancement of ESD surgery. We hope that this explanation provides sufficient clarity on our dataset selection and can help build trust in the data and the accuracy of the model. We appreciate the reviewer’s feedback and have provided additional details in Page 2, lines 102-112 of our revised manuscript to address these concerns:

“The included cases cover a wide variability of lesion sizes, locations (i.e., rectum, stomach and esophagus) and surgical tools (i.e., dual / isolation-tipped / triangle-tipped knife). More details about the variability of the dataset are provided in Supplementary Table 1. Although the dataset spans a long time of 12 years, for the whole period the endoscopist has already achieved the level of expertise. At the start point (July 2008) of data collection, the endoscopist had conducted more than 100 ESD cases on each organ of rectum, stomach, and esophagus. According to the learning curve reported in [34, 35, 36], the endoscopist can be treated as an expert because the numbers of conducted cases are higher than the threshold (80/30/30 cases of rectum/stomach/esophagus, respectively).”

Figure 1: The examples of start and end frames of phases Marking, Injection, and Dissection at 1 fps. (Supplementary Figure 3)

Reviewer Comment 1.2 — “Annotation protocol: how did you define phase transitions, e.g. the start and end frame exactly”

Phase	Image Example	Annotation Protocol
Marking		 Surgical knives, including triangle-tipped knife, insulation-tipped knife and dual knife, get close to the tumor and place the knife onto the surface of mucosa; Coagulation marks are labelled around 2mm apart and 5mm vertical to the tumor; Start frame: Surgical knife is placed on the surface of mucosa; End frame: Surgical knife is retrieved or not in contact with mucosa for more than 3 consecutive frames.
Injection		 The injection needle is injected into the working channel, and the tip of tool is exposed and inserted into the submucosal layer; After sufficient solutions, e.g., hyaluronic acid and dextrose water, are injected from the probe into the mucosa, the injection tool is retracted; The tumor is lifted for better observation; Start frame: Injection needle is inserted into the submucosal layer; End frame: Injection needle is retrieved from the submucosal layer or not in contact with submucosal layer for more than 3 consecutive frames.
Dissection		 The surgeon sets the electro-surgical unit (ESU) and make sure the cutting surface of the knife (e.g., surgical knives, including triangle-tipped knife, insulation-tipped knife and dual knife) is exposed; The surgeon controls the knife to cut around the circumference of the tumor, during which injection might be repeated to keep enough elevation; After the dissection is completed, the lesion is removed with graspers; Start frame: The knife is used to cut the circumference of the tumor; End frame: The knife stops cutting or is not in contact with the soft tissue for more than 3 consecutive frames.
Idle		 Phase Idle is defined as the hold-on time for the endoscopist to change the surgical tools, adjust the angle of scope and clean the field of view; The time used by the surgeon for inspecting the status of procedure and making decisions on following procedures; Start frame: When surgical tool has been retrieved or hovered the tool for more than 3 consecutive frames; End frame: Any one of phases Marking, Injection, and Dissection begins.

Figure 2: The annotation protocol of surgical phases Marking, Injection, Dissection and Idle. (Figure 1a in the manuscript)

Reply: Thank you for your comment on this important point. We in fact had careful considerations and multiple rounds of discussions between engineering and clinical team members to decide the annotation protocol including the phase definitions and their transition time-points with clear definitions.

The start and end frames are determined based on how the surgeons manipulate the surgical tools which are interacting with the soft tissues (Figure 1). Our definitions of ESD phases can be referred to the descriptions of functional tools provided in [4]. In Figure 2 (we also updated in the revised manuscript), we have added two bullets to clearly present how we determined the start and end frames for each phase. Specifically, 1) the start frame of Marking phase is when the triangle-tipped/insulation-tipped/dual knife is placed on the surface of the mucosa, and the end frame is when the knife is retrieved or not in contact with the mucosa for more than 3 consecutive frames; 2) For the Injection phase, the start frame is when the injection needle is inserted into the soft

Figure 3: The workflow of phase annotation. Blue lines represent the stages we prepared data for annotation, and yellow lines denote the annotation process. (Supplementary Figure 1)

tissue of submucosal layer, and the end frame is when the injection needle is retrieved from the submucosal layer or not in contact with the submucosal layer for more than 3 consecutive frames; 3) For the Dissection phase, the start frame is when the triangle-tipped/insulation-tipped/dual knife is used to cut the circumference of the tumor, and the end frame is when the knife stops cutting or is not in contact with the soft tissue for more than 3 consecutive frames; 4) For the Idle frame, the start frame is when any surgical tool has been retrieved or surgeons hover the tool without doing any action for more than 3 consecutive frames, and the end frame is when any of the other phases begins.

To facilitate the annotation process, we designed a data flow to determine the start and end frames (see Figure 3). Due to the large volume of the dataset, we first downsampled the video to 1 fps and saved it as a sequence of consecutive images. An Excel file was then generated to record the phase annotation frame-by-frame. During the annotation procedure, the annotators primarily focused on the downsampled images to identify the start and end frames of each phase, and subsequently labeled all frames within this duration as corresponding to the designated phase. Additionally, the annotators referred to the raw video to gain insight into the visual cues. Visual cues can be helpful for annotators in discriminating between different surgical phases during a procedure. For example, during the Marking phase, the endoscopist may inspect the contact point between the knife and mucosa. During the Injection phase, the endoscopist may observe the movement and diffusion of the injected agent through the tissue. During the Dissection phase, the endoscopist may look for changes in tissue color or texture that indicate the separation of different tissue layers. This temporal information was critical for the annotators to confirm whether surgical tools were hovering in the air, i.e., the ending of a phase.

In our revised manuscript, we have made several updates. Firstly, we have included specific definitions for the start and end frames of each phase in Figure 1a. Additionally, we have added a Supplementary Note 1 to provide a detailed account of our annotation process, which includes

Figure 4: Annotation examples of two raters. (Supplementary Figure 2)

a description of the annotation workflow in “*Dataflow of large-scale phase annotation*” and a demonstration of our annotation results in “*Annotation results*”. We encourage the reviewer to refer to the “*Dataset Annotation*” section in the Supplementary material for further details.

Reviewer Comment 1.3 — “How were annotation disagreements during the verifications process amongst the ESD surgeons resolved? How often did this occur, e.g. what was the inter-rater agreement?”

Reply: Thank you for the constructive comment.

We have two trained annotators to initially label the data, and two experienced endoscopists to review all the labeled data together. In the case of any discrepancies, the two experienced endoscopists discussed for a consensus label, while with more respect to the relatively senior one who has conducted more ESD cases. Our entire annotation process was as follows:

- 1) First, we randomly selected approximately 10% (5 cases, 20,446 frames) of the expert dataset to be annotated by two annotators. Following the predefined protocols, the two annotators individually annotated the selected data using the data flow outlined in Figure 3. This step not only helped the annotators become familiar with the annotation workflow, but also served as a quality control measure for the annotations. We used the Pearson correlation coefficient (PCC) [5] to quantitatively evaluate the interobserver agreement between the two raters. The PCC for the selected samples was 0.93, indicating a high level of agreement between the raters. To examine the disagreements visually, we display the phase annotation examples from the two annotators in Figure 4. For the majority of the example, the annotations of two raters are nearly identical, with only a small percentage (3.30%) exhibiting ambiguity between the annotations of different raters.
- 2) Considering the high level of consistency in the initial annotation results of the two annotators, we divided all annotation tasks into approximately equal halves, with each annotator individually annotating one part. Taking the 47 cases of expert dataset as an example, the dataset was first divided into two non-overlapping parts, including 24 and 23 cases respectively. Then, two annotators annotated the 24 cases (108,286 frames) and 23 cases (92,740 frames) individually. Given the high consistency between annotators, we collected these two groups of annotation and treated them as the final annotation of expert dataset. We employed the same strategy to distribute the datasets to raters for the external, ex-vivo and in-vivo datasets. Specifically, one rater annotated 8 cases (66,756 frames), 2 cases (18,459 frames) and 6 cases (12,990 frames) of external, ex-vivo, and in-vivo datasets

respectively. For another rater, 7 cases (55,412 frames), 2 cases (11,655 frames), and 6 cases (16,512 frames) were annotated respectively.

- 3) After the two annotators completed all of their annotation tasks, the annotations underwent quality control by two experienced endoscopists with six and three years of experience respectively. Two endoscopists reviewed the annotations with a focus on correcting challenging annotations that even trained annotators may struggle with. The endoscopists relied on not only visual cues but also practical surgical experience to determine the appropriate surgical phase in situations where the surgical site was highly complex or key anatomical landmarks were obscured. To assist the endoscopists during the verification process, we provided synchronized information by overlaying the annotation results from the second step onto the raw video. The endoscopists then quickly reviewed the video and corrected any errors as needed. This step yielded our final phase annotation dataset.

In our revised manuscript (page 3, lines 25-43), we provide more detailed information on the specific procedures employed to resolve annotation disagreements and the inter-rater agreement metrics obtained.

“To ensure high-quality data, the annotation workflow consisted of three stages. Firstly, two well-trained medical annotators (Y.Y. Chen and M.K. Cheng) independently annotated approximately 10% (5 cases, 20,446 frames) of the expert dataset based on the dataflow in Supplementary Figure 1. Then inter-rater agreement was measured using the Pearson correlation coefficient (PCC) [39], which ultimately reached 0.93. Supplementary Figure 2 provides an overview of the annotations from the two raters. Secondly, the two raters collaborated to annotate all datasets by dividing all annotation tasks into approximately equal halves, with each rater individually annotating one part. Finally, after the two annotators completed all of their annotation tasks, the annotations underwent quality control by two experienced endoscopists. The endoscopists relied on not only visual cues but also practical surgical experience to determine the appropriate surgical phase in situations where the surgical site was highly complex or key anatomical landmarks were obscured. Details on the dataflow, annotation schedule, and annotation results are provided as Supplementary Note 1.”

Additionally, to facilitate understanding of the dataset formulation, we describe the three-step annotation procedure in our Supplementary material as a subsection of “*Dataset annotation*”. The reviewer may refer to the Supplementary for more details.

Reviewer Comment 1.4 — “While the authors go in great detail in regards to the different metrics for the different phases for the data of the expert surgeon, these details are missing in the evaluation of the external dataset, the ex-vivo and the in-vivo porcine study. To allow for a thorough evaluation and comparison of the performance in the different domains, a table similar to 2b) (or its contents) should be provided for each domain. As the different classes are highly imbalanced, overall accuracy does not give a complete picture.”

Reply: We appreciate the reviewer’s comment regarding the evaluation of the external dataset, ex-vivo, and in-vivo porcine study. We agree that providing detailed metrics for each phase is

important to facilitate a thorough evaluation and comparison of the performance in the different domains.

Table 2: Performance Metrics on External Dataset. (Supplementary Table 2)

Phase-wise metric	AUROC	Specificity	Sensitivity	Orderliness
Marking	97.30 (94.45, 100.0)	93.23 (88.35, 98.12)	94.09 (89.79, 98.40)	93.23 (88.36, 98.11)
Injection	98.87 (98.06, 99.68)	96.53 (95.19, 97.87)	96.69 (95.16, 98.23)	96.53 (95.20, 97.86)
Dissection	95.69 (92.34, 99.05)	93.10 (91.01, 95.19)	90.16 (85.96, 94.36)	92.36 (90.47, 94.24)
Idle	96.68 (95.82, 97.55)	90.82 (88.55, 93.09)	92.07 (90.41, 93.73)	91.67 (90.11, 93.22)
Average	97.14 (95.17, 99.07)	93.42 (90.78, 96.07)	93.25 (90.33, 96.18)	93.20 (91.04, 95.86)

Table 3: Performance Metrics on Ex-vivo Dataset. (Supplementary Table 3)

Phase-wise metric	AUROC	Specificity	Sensitivity	Orderliness
Marking	57.87 (0.000, 100.0)	35.65 (0.000, 100.0)	86.17 (53.94, 100.0)	36.23 (0.000, 100.0)
Injection	96.97 (92.16, 100.0)	94.11 (80.66, 100.0)	96.44 (91.64, 100.0)	94.14 (81.01, 100.0)
Dissection	95.67 (90.65, 100.0)	89.32 (82.48, 96.15)	90.81 (83.64, 97.98)	90.34 (83.47, 97.21)
Idle	94.74 (88.68, 100.0)	88.13 (78.03, 98.22)	90.13 (84.09, 96.18)	89.08 (81.19, 96.96)
Average	86.31 (67.87, 100.0)	76.80 (60.29, 98.34)	90.89 (78.33, 98.54)	77.45 (61.42, 98.54)

Table 4: Performance Metrics on In-vivo Dataset. (Supplementary Table 4)

Phase-wise metric	AUROC	Specificity	Sensitivity	Orderliness
Marking	69.02 (56.35, 81.69)	53.98 (39.08, 68.89)	90.58 (83.52, 97.64)	54.61 (39.93, 69.29)
Injection	96.58 (95.03, 98.12)	91.64 (89.11, 94.17)	91.82 (86.51, 97.13)	91.72 (89.43, 94.00)
Dissection	94.93 (92.97, 96.89)	91.57 (89.89, 93.24)	86.68 (83.22, 90.14)	88.95 (87.11, 90.79)
Idle	93.27 (92.12, 94.43)	81.62 (78.62, 84.63)	91.24 (89.75, 92.72)	85.47 (83.63, 87.31)
Average	88.45 (84.12, 92.78)	79.70 (74.18, 85.23)	90.08 (85.75, 94.41)	80.19 (75.03, 85.35)

To address this concern, we have now provided comprehensive performance metrics for the external dataset, ex-vivo, and in-vivo animal study separately, similar to the approach taken for the data of the expert surgeon. We include Tables 2, 3, and 4 similar to Figure 2b that summarize phase-wise performance metrics for each dataset, including specificity, sensitivity, area under ROC curve (AUROC), and orderliness. This allows for a more informative comparison of performance across different domains.

As demonstrated in tables 2, 3, and 4, the AI-Endo consistently achieves high performance on the Injection and Dissection phases across all three datasets. For clinical surgery cases in the external dataset, in which surgeries were conducted by different endoscopists with varying skill levels, the AI-Endo maintains an overall AUROC of 97.14 (CI: 95.17, 99.09), demonstrating its potential for clinical applications. A performance decline exists during the Marking phase in ex-vivo and in-vivo datasets. The Marking phase exhibited high sensitivity but low specificity, indicating that AI-Endo was able to accurately identify non-marking frames but struggled to precisely locate marking frames. This decrease in performance may be attributed to a shift in data distribution between clinical and animal trial datasets, such as differences in blood vessel

characteristics. Besides, the short duration of contact between the tissue and marking tool can result in a limited number of training examples and make the Marking phase particularly challenging to be recognized.

We thank the reviewer for highlighting this important aspect, and we are committed to addressing this comment by providing comprehensive performance metrics. For the sake of smoother content flow and space limitation, we have included tables 2, 3, and 4 in the supplementary material. Additionally, we have explicitly referred to these tables in the main text for a comprehensive understanding of the model’s performance (Page 5, lines 34-35; Page 6, lines 53-54; Page 7, lines 26-27).

Reviewer Comment 1.5 — “Was the same annotation protocol follow for the external dataset, the ex-vivo, and the in-vivo porcine study as was for the expert dataset? This is not entirely clear in the paper. ”

Reply: Yes, in our work, we followed the same annotation protocol for the external, ex-vivo, and in-vivo datasets as used for the expert dataset. We thank the reviewer for the feedback regarding the annotation protocol for the external dataset, ex-vivo, and in-vivo animal study. It is important to ensure consistency in the annotation protocol across all datasets to facilitate a fair comparison of the model’s performance. The annotation process on all datasets strictly followed the dataflow in Supplementary Note 1. Specifically, we downsampled all endoscopy videos and saved them as consecutive images at 1 fps. Annotators then annotated the phase of each frame by checking both the downsampled images and the raw video simultaneously, following pre-defined protocols.

We apologize for the lack of clarity in annotation procedures outlined in our manuscript. In our revised version, we have made it explicit that the same annotation protocol was applied consistently to all datasets, as can be found in Page 3, lines 51-53:

“The same annotation protocol and dataflow were applied to expert, external, ex-vivo, and in-vivo datasets.”

Reviewer Comment 1.6 — “On what data were the models for the external dataset, the ex-vivo and the in-vivo porcine study trained? Was the entire expert dataset used or was one of the 5 models used for the expert cross-validation-study used?”

Reply: We appreciate the reviewer’s question regarding the training data used for the models in the external, ex-vivo, and in-vivo porcine study. In our work, we used the entire expert dataset (47 cases) to train the model and then used this model for evaluations on other datasets, including external, ex-vivo, and in-vivo datasets.

Specifically, we fine-tuned the hyper-parameters of deep learning model in cross-validation-study and evaluated the performance of trained model with the five folds. Given the hyper-parameters in the cross-fold evaluation, we retrained the model with the entire developmental dataset to maximize the amount of available training data and improve the performance of the model [6]. For the applications of AI-Endo, such as evaluation on external datasets, mutli-center

datasets, ex-vivo and in-vivo animal studies, we utilized the model trained on the entire expert dataset with 47 cases.

We thank the reviewer for the valuable comment, and we have clarified this point in our revised manuscript:

- a) We have added the statement, “...all cases of the developmental dataset were first arranged in chronological order, and then we sampled them at five equal intervals. This procedure resulted in 5 folds...” (Page 11, lines 14-16).
- b) We have added the statement, “...After optimizing and fixing the network structure and hyper-parameters, we proceeded to retrain the model using the entire developmental dataset. This was done to maximize the amount of available training data and improve the model's generalization performance. For any future applications of AI-Endo on other datasets, such as external, ex-vivo, and in-vivo animal studies, we used the model that was trained on the entire developmental dataset.....” (Page 11, lines 37-44).

We hope that these revisions address the concerns raised by the reviewer and provide a more comprehensive and transparent explanation of our training process.

Reviewer Comment 1.7 — “How were the final models selected? Did you use the model that resulted after the 4000 iterations of the 2nd stage?”

Reply: We are grateful to the reviewer for the comment and for the opportunity to clarify our model selection process. Our final model was chosen at 4000 iterations based on its performance during the second stage of our training process. To ensure the reliability of our model selection, we have included Figure 5 in the supplementary materials, which displays the validation loss curves corresponding to the two stages of our training process in our preliminary experiments. We observed that the validation loss seemed to be stable around 8000 and 4000 iterations respectively, therefore we halted the training processes and use the last iteration as our final training result.

We would like to emphasize that our model training process involved several stages and components. We adopted a two-stage training approach to reduce the demand for computational resources. In the first stage, we trained and selected the final ResNet50 module, which remained fixed throughout the subsequent stages. Prior to starting the second training stage, we extracted frame-wise feature embeddings using the ResNet50 module and cached them. During the second stage, we loaded these features and processed them using the Fusion and Transformer modules, whose parameters would be optimized at this stage. Our Fusion module utilized temporal convolution module to extract casual features [7], enabling the model to causally process all consecutive frames of the video simultaneously. We also tried to train the ResNet50, Fusion, and Transformer modules end-to-end, but the ResNet50 struggled to process long sequences due to limited memory (NVIDIA 3090, 24G). During inference, frames of the video stream were sequentially input into the ResNet50, Fusion, and Transformer modules, yielding the final predictions.

In our revised manuscript, we have provided details on the model selection process. We included the following statements: “When the first stage of training was finished, we fixed and utilized the trained model...” (Page 11, Line 24); “By adopting temporal convolution [57], the

Figure 5: The curves of the training loss of a) ResNet50 module in the 1st stage and b) Fusion and Transformer modules in the 2nd stage. The dashed line indicates the number of iterations we actually trained. (Supplementary Figure 5)

model was empowered to process all feature embeddings of the video in a causal manner, thus preserving the characteristics necessary for online prediction.” (Page 11, lines 28-31); “Supplementary Figure 5 shows the curves of training loss, where the loss curve became flat at the end of the training process. Therefore, models at the final iteration were used for phase prediction” (Page 11, lines 34-37) to address the training process and provide additional insights into the model selection.

Reviewer Comment 1.8 — “Could you elaborate further on your ‘*orderliness*’ metric? To me it seems identical to accuracy (assuming $\hat{T}F$ is supposed to be $\hat{T}P$) and I don’t understand how it captures the order of the frames per phase”

Reply: We would like to express our gratitude to the reviewer for bringing to our attention the confusion related to the *orderliness* metric and for helping us to improve its description. The orderliness metric is derived from the Youden index’s optimal cut-off and measures the degree of how the target phase is *ordered* after the other phases in the probability scale bar. We named this metric orderliness to differentiate it from the commonly used accuracy metric, which employs a fixed threshold of 0.5. The optimal threshold of the Youden index is determined by maximizing the summarization of sensitivity and specificity [8], resulting in a more accurate measurement of how well the positive and negative samples are separated relative to the optimal threshold on the probability scale bar [0,1]. The Youden index has been extensively discussed in previous research as a means of characterizing the class-wise performance of a model on a multi-class classification problem [9,10]. The use of “ $\hat{T}F$ ” was a typo, and it should be “ $\hat{T}P$ ” to represent the number of true positive samples under the specified cut-off.

To be more specifically, the model outputs the probability of each frame belonging to one of four distinct phases, Marking, Injection, Dissection, and Idle. This allows us to evaluate the model’s performance both overall and for each individual phase. To help clarify the phase-

wise metric orderliness presented in Figure 2b, we use the example of the Dissection phase to illustrate how orderliness describes the ratio of Dissection frames that are correctly *ordered* after the non-Dissection frames in the probability bar. We begin by taking the probability of each frame x being Dissection, denoted as $p(\text{Dissection}|x)$. Using the `sklearn.metrics.roc_curve` package, we generated a ROC curve that plots the true positive rate (TPR) against the false positive rate (FPR) for thresholds uniformly sampled in the range $[0, 1]$. We then calculated the optimal threshold $t^* = 0.4$ from the Youden Index, $J_{\max} = \max_t \text{sensitivity} + \text{specificity} - 1 = TPR + (1 - FPR) - 1 = TPR - FPR$. With the optimal threshold t^* , we divided the samples into four groups: $\hat{T}P$, $\hat{T}N$, $\hat{F}P$, and $\hat{F}N$. Finally, we defined $\text{orderliness} = \frac{\hat{T}P + \hat{T}N}{\hat{T}P + \hat{T}N + \hat{F}P + \hat{F}N}$ to measure the proportion of frames that were correctly sorted relative to the threshold t^* .

We visualized the steps involved in calculating the orderliness metric of phase Dissection in Figure 6. In Figure 6a, we obtained the optimized threshold t^* from the ROC, and the inset illustrates the specific process we used to calculate orderliness, as described above. Additionally, we have depicted the relationship between the threshold t^* and the probability distributions of Dissection and non-Dissection frames in Figure 6b. Ideally, t^* would be able to perfectly discriminate between the boundaries of these two distributions, which means the output probability of all Dissection frames should be higher than that of non-Dissection frames. However, the AI model may fail to recognize challenging cases, such as a blurry view or obscured surgical tools. This can result in prediction errors, i.e., $\hat{F}P$ and $\hat{F}N$, consequently dividing all samples into four groups, as shown in Figure 6b. To assess the performance of the AI model with respect to each phase, we define the orderliness to calculate the proportion of frames that are correctly classified, i.e., $\hat{T}P$ and $\hat{T}N$.

Figure 6: Definition of metrics *orderliness*; a). The steps for calculating *orderliness*; b). The distributions of output probabilities corresponding to Dissection (in red) and non-Dissection (in purple) frames. (Supplementary Figure 4)

We thank the reviewer for helping us refine the description of orderliness metric. In the revised manuscript, we have detailed the definition of orderliness with the example of Dissection:

“Taking the Dissection phase as an example, the output probability of Dissection frames being predicted as Dissection should be higher than that of non-Dissection frames, which means the larger the number of Dissection frames ordered at the higher range of probability $[0, 1]$ is, the better the performance of the model will be. Details on the orderliness metric are provided in Supplementary Note 2.” (Page 4, lines 60-67).

Besides, we have added Supplementary Note 2 into the Supplementary material to discuss definition and usefulness of orderliness metric. The Figure 6 is also included in the Supplementary material to visually demonstrate the process of orderliness calculation and its relationship with the output probability distributions.

Reviewer Comment 1.9 — “Could you comment on the difference in the percentage of the duration spent in ‘idle’ in expert vs. external dataset?”

Reply: We thank the reviewer for raising this question and providing additional insights into the difference in the percentage of duration spent in the “Idle” phase between the expert and external datasets. This helped us refine our explanation and provide a more comprehensive understanding of the differences observed between the expert and external datasets.

Considering the annotation protocol and the variations in surgical proficiency, it is expected that the external dataset, which includes recordings from one experienced and two junior endoscopists, would exhibit a higher ratio of the “Idle” phase compared to the expert dataset consisting of one expert endoscopist. This observation aligns with our findings, where the percentages of “Idle” phases were 0.547 (CI: 0.467, 0.628) and 0.349 (CI: 0.322, 0.377) in the external and expert datasets, respectively.

As the reviewer correctly mentioned, the expertise and experience of the endoscopist play a significant role in the continuity of surgical operations. Studies such as [11] and [12] have highlighted that junior surgeons may exhibit longer durations for tool changes or decision-making, resulting in increased instances of hesitation or tool retrieval, which are categorized as “Idle” phase according to our annotation protocol.

In the revised manuscript, we have incorporated this explanation in description of expert dataset as follows (Page 2, lines 112-115):

“Compared to the external dataset, the low percentage of phase Idle (0.349 (CI: 0.322, 0.377) vs 0.547 (CI: 0.467, 0.628)) further confirms the high expertise level of the developmental dataset [36], [37].”

Reviewer Comment 1.10 — “In section ‘Statistical analysis’ you state that a significance analysis was performed though I cannot seem to find it in the text.”

Reply: We thank the reviewer for bringing attention to the significance analysis mentioned in the manuscript’s “Statistical Analysis” section. In accordance with the *Guide for Submission to Nature Communications*, we included this statistics section to state the methods we used in our work. Specific values were inserted in the corresponding parts of section *Results*.

“All statistical analyses were performed with Python (v3.6). For the quantitative results of the performance on the development and external datasets, we adapted Student’s t-distribution with 95% confidence interval (CI: lower%, upper%). To compare the analytical results from different groups, we used two-sided pairwise T-test to inspect their statistical difference. P-value of < .05 was considered as statistically significant.”

In reporting our performance metrics, we used a 95% confidence interval denoted as num% (CI: lower%, upper%). For example, we illustrated the phase-wise performance in Figure 2b, reported the surgeon-wise accuracy on Page 5, lines 34-37, and the overall accuracy in the in-vivo animal study on Page 7, lines 28-29.

To determine significant differences between groups, we used two-sided t-test and considered two groups to be significantly different if the resulting p-value was less than 0.05. For instance, we observed a significant difference in the Normalized Transition index between experienced and novice endoscopists (Page 7, lines 86-88).

Reviewer Comment 1.11 — “What do the photos in 5a) show?”

Reply: We thank the reviewer for pointing this out. The photos in Figure 5a) show the dissected samples collected in the animal trials. In the in-vivo animal trial, we took a photo for each sample with the ruler placed aside. The size information could be used for statistical analysis, e.g., skill assessment.

Figure 7: Data analytical results derived from AI-Endo phase recognition for in-vivo animal experiments. (a) The curves of derived online score of Normalized Transition index calculated for the senior surgeon (2 cases with orange lines) and the novice (2 cases with brown lines) at esophagus (Inserted photos represent the dissected samples). Scale bar, 1cm; (b) Design of the AI summary report that is automatically generated by the AI-Endo system in pre-clinical trial. See more detailed explanations in the main text.

In our revised manuscript, we have added “Inserted photos represent the dissected samples”

in the caption of Figure 5a).

Reviewer Comment 1.12 — “Do you plan to make the datasets publicly available? I would expect this for a high-impact publication as contribution and it would be highly appreciated by the community”

Reply: We fully agree with the reviewer that public datasets can advance research in this field, and we are pleased to make the ex-vivo and in-vivo datasets, as well as the source code of AI-Endo, publicly available upon the publication of this work. However, considering the confidentiality and privacy concerns regarding medical data, the developmental and external cases involving human subjects cannot be made publicly accessible at this time point. We are applying for IRB from the university ethics committee for data release.

In the revised manuscript, we have added the statement in section “*Data Availability*” as:

“The in-vivo and ex-vivo animal trial data are available in the data repository at [57]. However, due to ethical regulations on confidentiality and privacy concerns, access to the human cases in the developmental and external datasets of this study is limited to authorized researchers approved by the ethics committee.”

Response to reviewer 2

General Comment — “The manuscript entitled ‘Intelligent Surgical Workflow Recognition for Endoscopic Submucosal Dissection with Real-time Animal Study’, you tried to elucidate promising performance of AI-Endo, an intelligent surgical workflow recognition suit, for AI assistance in endoscopic submucosal dissection (ESD).”

Reply: We would like to express our gratitude to the reviewer for your summary of our work. In our initial submission, we proposed an AI-powered framework, called AI-Endo, for real-time recognition of the surgical phase of ESD surgery. In contrast to previous works, we conducted both in-vivo and ex-vivo animal studies to further validate the reliability of our model in practical applications. Our findings clarify the tremendous potential of intelligent algorithms, particularly deep learning, in surgical scenarios.

We greatly appreciate the reviewer’s efforts in guiding us to improve our work, particularly with respect to the application of AI-Endo on different endoscopy equipments (Fujifilm and Pentax systems) and techniques (pocket creation, traction, and water injection). In this revised version, we have collected new surgical cases from our collaborators to address these questions. Firstly, we obtained three cases of pocket creation and one case of line-assisted traction from Prince of Wales Hospital, Hong Kong to validate AI-Endo’s capability in recognizing surgical phases during these specific procedures. AI-Endo achieved accuracies of 93.07% (CI: 83.44%, 100.0%) on pocket creation and 75.22% on traction. The decline in performance largely depends on whether specialized tools are used in the procedure; otherwise, AI-Endo retains the capability of recognizing surgical phases involved in these techniques. Secondly, we obtained four cases of Fujifilm Endoscopy from Nanfang Hospital, Southern Medical University, Guangzhou, China, on which AI-Endo achieved an average accuracy of 90.75% (CI: 88.50%, 93.01%), indicating its potential to be applied to different endoscopy systems. Thirdly, we collected four cases from Hospital of Augsburg’s Internal Medicine III - Gastroenterology, Germany to establish a international multi-center dataset. AI-Endo obtained an average of 87.34% (CI: 84.43%, 90.25%), showing its applicability to different centers under regional variance. In the revised manuscript, we have enriched the section “*Performance of AI-Endo model on external datasets*” with validation on novel techniques. Besides, we have added a section “*International multi-center external validation*” to explicitly discuss applicability of AI-Endo to different endoscopy systems and multi-center dataset.

Additionally, we discussed the details of our annotation protocol. In our study, we did not divide the Dissection phase into mucosal incision and submucosal dissection due to the blurry boundary and similar functions between these two sub-phases. For the Marking phase, we kept its annotation because the continuity and duration of this phase may contribute to surgical skill assessment, even though it only occurs at the beginning of the surgery. Besides, we clarified the endpoints of our ex-vivo and in-vivo studies. The ex-vivo study was designed to ensure that AI-assistance could deliver useful data analytic results and reduce interruptions caused by add-on AI functionality in a cost-effective manner. On the other hand, the in-vivo study aimed to showcase the applicability of the intelligent phase recognition system with online score analysis

and automatic performance reports.

We kindly recommend the reviewer to refer to the following parts, in which we have responded to the comments point-by-point with more details. We believe that this revised manuscript will better convey our research target and demonstrate the capabilities of AI-Endo.

Reviewer Comment 2.1 —

“Development dataset and external validation dataset were obtained by human cases and validation of AI-Endo was only made by ex-vivo and in-vivo animal models. Further study by using human cases is warranted to show the real performance of AI-Endo.”

Reply: We thank the reviewer for the valuable feedback and appreciate the suggestion to validate the model on human cases through clinical trials. We understand the importance of clinical trials in validating the performance of AI-Endo, and is currently applying for ethics approval for a human clinical trial in Hong Kong. The results will be included in our future publications. Fortunately, in this study, we successfully validated the prediction accuracy and computational efficiency of AI-Endo using an external dataset of recorded human cases and animal trials. We achieved a promising accuracy of 91.48% and an efficiency of 47 frames per second, indicating the great potential of AI-Endo in intraoperative settings. This brings us strong confidence in successfully deploy AI-Endo on the human case in our future work.

In our *Discussion* section (Page 10, lines 16-19), we have provided our plans for future clinical studies.

“We aim to include clinical trials in our future work after the entire system is extensively validated with more surgeons and clinical centers, ensuring participant safety in invasive procedures.”

Reviewer Comment 2.2 — “Primary endpoints for a series of pre-clinical studies are unclear. What is the possible intervention by using AI-Endo in clinical use? Did AI-Endo evaluate only the performance of endoscopists?”

Reply: We thank the reviewer for helping us clarify the possible intervention and practical usage of AI-Endo. Here we would like to answer this comment from two aspects: primary endpoints and potential usage.

1. Primary endpoints of pre-clinical studies

The pre-clinical studies in this work consist of ex-vivo and in-vivo animal trials, with the following endpoints for each study:

- 1) The ex-vivo animal trial focused on testing the computation efficiency and accuracy of AI-Endo when integrated into the endoscopy system. Because existing literature on computer-assisted surgery in general have not yet clearly investigated how to properly incorporate the AI-Endo model into current clinical workflow, we needed ex-vivo animal study to optimize and validate the proposed framework in our work, ranging from the layout of third-party

monitors to the design of the graphical user interface. Compared to in-vivo animal study, it is more cost-effective to ensure the AI-assistance not only delivers useful data analytic results but also avoids interruptions caused by add-on AI functionality.

- 2) The in-vivo animal trial was designed to exemplify the practical usage of AI-Endo in surgical training. Specifically, we conducted a controlled experiment with two endoscopists at different experience levels performing ESD surgery. Based on the automatic phase recognition results, we proposed the Normalized Transition index to compare the endoscopic skills during training sessions and finally generated performance reports on procedure duration and fluency.

In the revised manuscript, we have now explicitly specified the endpoints of ex-vivo and in-vivo animal trials to help readers understand the significance of ex-vivo and in-vivo animal trials. Specifically, we made changes as follows:

- Add a summary on pre-clinical studies at the end of Section *Introduction* (Page 2, lines 68-75). *“To evaluate the computational efficiency and compatibility of AI-Endo in real-time applications, we first conducted cost-effective ex-vivo animal trial using video streamed from an endoscopy (Olympus Medical Corporation, Tokyo, Japan) to our AI workstation. Thereafter, we designed an in-vivo animal study to showcase the potential of AI-Endo in surgical training, including the generation of performance reports and the comparison of surgical skills.”*
- We explained the endpoint of ex-vivo study at (Page 6, lines 12-22):
“Existing works on surgical phase recognition have not yet clearly investigated the incorporation of AI-Endo model into clinical workflow, therefore, we designed ex-vivo animal study to optimize and validate the proposed framework in our work, ranging from the layout of third-party monitors to the design of the graphical user interface. Compared to conducting in-vivo animal study directly, adopting a preliminary ex-vivo study first is more cost-effective to ensure the AI-assistance could deliver useful data analytic results and alleviate interruptions caused by add-on AI functionality. To confirm how to seamlessly integrate the AI-Endo computational tool into the Endoscopy System, we implemented the whole system in a training laboratory at CUHK Jockey Club Minimally Invasive Surgical Skills Centre.”
- We revised the start part of in-vivo animal study to discuss the endpoint of in-vivo study, which can be found on (Page 6, lines 60-62; Page 7, lines 1-5):
“Quantities of works have been proposed for automated surgical phase recognition, however, none of them incorporated in-vivo animal trials to demonstrate the clinical application of system in real-world surgery. Based on the success of ex-vivo animal experiments, we further conducted an ESD surgical training session with in-vivo live animal trials, aiming to showcase the clinical applicability of intelligent phase recognition system with online score analysis and automatic performance report generation.”

2. Potential usage of AI-Endo

Based on automatic phase recognition results, AI-Endo generated analytical report on the surgical process in real-time. In our work, we demonstrate the significant potential of AI-Endo in addressing a widely recognized challenge in skill assessment. AI-Endo's real-time performance enables online assessment of surgical performance, as demonstrated by the Normalized Transition index in the in-vivo animal study. Although a single score cannot reflect the surgical level of endoscopists comprehensively, AI-Endo's robust performance across different datasets allows researchers to explore other scores based on phase recognition results.

Additionally, surgical phase recognition is a fundamental task in a series of AI-assisted surgical applications. For instance, AI-Endo could potentially request remote assistance when the procedure enters a critical stage, e.g., a junior surgeon begins dissection. To assist with surgical quality improvement [13], AI-Endo enables the generation of structured and segmented databases corresponding to specific phases. Moreover, by accurately identifying important operations, phase recognition could serve as a groundwork for more complex tasks in AI-assisted surgery, such as tissue segmentation and dissection trajectory planning.

In the revised manuscript, we have added a description about the application scenarios of automatic phase recognition (Page 9, lines 64-70):

"The benefits of automatic phase recognition go beyond the generation of statistical reports and the calculation of online NT-index, which provide only a limited view of surgical skill evaluation. We encourage community researchers to utilize the open-source code and data we provide to explore the statistical significance of surgical phases and promote progress in surgical training and related areas, such as establishing large-scale structured and segmented surgical phase databases [54]."

Reviewer Comment 2.3 — "You divided a sequence of ESD into 4 phases, marking, injection, dissection, and Idle. However, marking is only made in the very initial phase, and the others are repeated during ESD. Usefulness of the classification would be confirmed before making annotation and training AI. "

Reply: We would like to express our appreciation to the reviewer for pointing out the necessity of annotating and recognizing phase Marking. While we agree that Marking typically occurs during the initial stage, endoscopists sometimes hesitate when locating the marking point, resulting in discontinuity of Marking phase. Previous studies have suggested that the continuity and fluency of surgical operations, including the Marking phase, can partially reflect the level of surgical skill [11]. Therefore, we decided to assign the frames of Marking as an individual class instead of Idle.

The Marking process of experts retains a high degree of continuity, leading to less repetition of Idle phase. To quantify this observation, we calculated the total duration and the number of marking phase segments in both the developmental dataset (one expert endoscopist) and the external dataset (three less experienced endoscopists). The results showed that the expert endoscopist has 1.21 segments and 2.27 minutes of total marking duration, while the three less

experienced endoscopists have 15.92 segments and 2.55 minutes of total marking duration. This likely reflected less interruption (Idle phase) during the marking phase by the expert endoscopists, confirming the potential utility of recognizing the Marking phase.

Reviewer Comment 2.4 — Injection phase may include not only injection with injection needle. But water injection with electrosurgical knives or the scope tip would be also injection phase.

Reply: We are grateful to the reviewer for bringing our attention to the issue of water injection in ESD surgery. Typically, water injection is used during ESD to enhance the submucosal cushion, and it only lasts for 1-2 seconds during submucosal dissection. The water injected during ESD is usually transparent and rapidly dispersed, making it difficult to distinguish from the surrounding tissues. Consequently, it is technically challenging to retrospectively review and annotate the frames with human eyes. To alleviate annotation difficulty and ensure accurate labeling, we did not explicitly include the identification of water injection and treated it as Dissection phase in our dataset. As a result, our trained model has limited ability to recognize water injection due to the short duration of the injection phase and the lack of training samples.

In practical terms, there may be no significant value in including water injection in the AI-Endo phase recognition. The primary objective of this system is to analyze the procedure for performance assessment or as a framework for further AI development, such as identifying safe zones during the Dissection phase. Therefore, it would be more desirable to focus on recognizing surgical operations that commonly occur in ESD procedures. For customized usage in specific procedures, we recommend collecting specific data and fine-tuning the AI-Endo model for optimal performance.

In the manuscript, we have improved the annotation protocol of Injection phase as (Page 3, lines 8-12):

“Injection: submucosal elevation would be achieved by injection of a mixture of solutions containing normal saline, epinephrine, hyaluronic acid using needle injector. Due to the difficulty in retrospective annotation and often ultra-short duration of 1-2 seconds, transient saline injection through the channel within the electrosurgical knives could not be separately annotated and thus would be included in the Dissection instead of Injection phase.”

Reviewer Comment 2.5 — “Dissection phase may be divided into mucosal incision phase and submucosal dissection in a narrow sense phase. In addition, how to classify images of bleeding or hemostasis is unclear. Better endoscopists can dissect less bleeding (red color images) and less use of hemostatic forceps (treatment of nonbleeding visible vessels by electrosurgical knives). In addition, better endoscopists would make less exchanges of devices and less proportion of idling time.”

Reply: We appreciate the reviewer’s comment on the definition of dissection phase and the classification of bleeding or hemostasis images. We agree that ESD procedures are complex,

demanding a careful design on the AI model to enable its application to clinical usage. Here, we would like to answer these questions point-by-point as follows:

1. Annotation of dissection phase

Specifically speaking, the dissection phase is composed of two stages, the mucosal incision phase and the submucosal dissection. During the mucosal incision phase, an initial incision is made in the mucosal layer surrounding the lesion. Meanwhile, in the submucosal dissection phase, the lesion is carefully separated from the submucosal layer beneath it. By utilizing this two-stage approach, the removal of the lesion is more efficient and safer, while the risk of perforation or bleeding is minimized [14].

However, from a practical standpoint, distinguishing between the mucosal incision and submucosal dissection stages can be challenging, even for experienced professionals. This is because the transition between the two stages can be gradual and blurry, making it difficult to define the boundary between them. In our work, we did not divide the dissection phase into mucosal incision and submucosal dissection for two reasons:

- 1) During the mucosal incision phase, the endoscopist creates an initial incision in the mucosa surrounding the lesion to be removed. However, the endoscopist may continue to dissect deeper into the submucosal layer during the initial incision (Trimming), blurring the boundary between the two phases;
- 2) The aim of AI-Endo is to recognize all frames that involve tissue dissection, during which adverse events such as bleeding are more likely to occur, especially for novice endoscopists. Because both mucosal incision and submucosal dissection involve dissection process, we combined them into one phase, i.e., Dissection phase, in this work.

2. Classification of bleeding or hemostasis images

We fully agree that the ability to recognize bleeding or hemostasis would be a highly valuable function in monitoring intraoperative surgical accidents, issuing alerts, and contributing to surgical skill assessment. Similar works have already been proposed for automatic bleeding detection in laparoscopic surgery [15, 16]. In our study, we focused on analyzing the surgical phases of ESD surgery and did not explicitly train the model for recognizing bleeding or hemostasis.

However, AI-Endo can correctly handle bleeding scenarios for phase recognition in two ways: 1) When only minor bleeding occurs, hemostasis could be easily achieved with electrosurgical knife coagulation, thus not requiring a change in endoscopic instruments. The time required to achieve such hemostasis is usually short and insignificant. Therefore AI-Endo could recognize it as Dissection phase. 2) When major bleeding occurs, the endoscopic view would be obscured by blood and hemostasis would likely require change into hemostatic forceps. All these interruptions would be interpreted as Idle phase by AI-Endo.

Expert endoscopists are often able to minimise intraprocedural bleeding through pre-emptive identification and coagulation of vessels, so the interruption during dissection phase would be minimised. As the reviewer noted, we observed a shorter idling time (24.45 minutes) for experts compared to less experienced endoscopists (71.14 minutes).

In the revised manuscript, we have refined the description about the definition on phase dissection in Section “*Annotation protocol of ESD workflow*” (Page 3, lines 14-19):

“Dissection (mucosal incision and submucosal dissection): mucosa around the marking point is incised, and then submucosal layer would be dissected from the underlying muscularis propria until the target lesion is resected and removed. Hemostasis with electrosurgical knives is included because of its short duration.”

Reviewer Comment 2.6 — “There are several novel techniques of ESD such as pocket creation method or traction method. Is AI-Endo applicable for these different techniques? ”

Reply: We thank the reviewer for raising the question of whether AI-Endo is suitable for novel ESD techniques, such as the pocket creation method and traction method. In regard to applying AI-Endo to these techniques, pocket creation and traction can be considered as two representative methods, depending on whether specialized tools are involved.

1. **Pocket creation method**

The pocket creation method improves visualization of correct dissection plane by creating a pocket in the submucosal layer [17]. This method typically involves four steps: 1) injecting saline solution near the lesion to create a space between the submucosal layer and muscle layer; 2) creating a small mucosal incision in the near side of the lesion to enter the submucosal space; 3) using dissection knives to create a pocket within the submucosa underneath the lesion and 4) Incision of the remaining mucosa to obtain en-bloc resection. The endoscopic instruments used in this method remain the same as those used in conventional approach. Although the pocket creation method is relatively new and not included in our developmental dataset, it comprises phases defined in the annotation protocol. Therefore, AI-Endo is expected to recognize surgical phases in this technique accurately.

To validate our statement, we collected three new cases of ESD performed with pocket creation method from Prince of Wales Hospital, Hong Kong. The procedures were strictly annotated according to the same protocol as our developmental dataset. Finally, we achieved an overall accuracy of 93.07% (CI: 83.44%, 100.0%) on the tested cases. AI-Endo retains its ability to recognize surgical phases included in the pocket creation process.

2. **Traction method**

The traction method during ESD involves the use of additional endoscopic devices such as a clip with a line, snare, or other commercially available traction devices to apply countertraction in facilitating dissection. This technique is particularly useful in some situations, such as esophageal ESD and gastric ESD in greater curvature, and when countertraction could not be adequately achieved by gravity during colorectal ESD [18]. Compared to what we have explored previously, the emergence of new functional tools makes it difficult for AI-Endo to accurately recognize surgical phases. As the surgical phase corresponding to the specified instrument is not included in the developmental datasets, AI-Endo tends to predict the frame as phase Idle.

Figure 8: Specific tool in ESD techniques. a) clip-with-line in traction; 2) hemostatic forceps in Idle phase.

We investigated this assumption with a new gastric ESD performed with line-assisted traction from Prince of Wales Hospital, Hong Kong. We evaluated the performance of AI-Endo on traction technique, which showed that AI-Endo obtained an accuracy of 75.22% (CI was not calculated for one case). When applied to techniques with specified tools, AI-Endo's performance is lower than the results obtained on the developmental dataset (91.04% (CI: 89.57%, 92.51%)). We visually inspected the results and found that the usage of new tools induced a decline in the ability of AI-Endo to understand phase-specific scenarios. For example, the clip-with-line used in this case is similar to the hemostatic forceps used in our study (Figure 8), causing AI-Endo to wrongly predict the traction process as Idle. This limitation coincides with our expectation because it is considerably difficult to include all specialized tools and novel ESD techniques over time in our developmental dataset, thus constraining the ability of AI-Endo in handling these situations.

We thank the reviewer for emphasizing the application of AI-Endo to novel ESD techniques. In the revised manuscript, we described the cases we used for validation on novel techniques (Page 4, lines 5-18) and the validation results (Page 5, lines 54-67):

- *"We also conducted further validation of our AI model in ESD procedures with novel techniques. Three ESD procedures of pocket creation method [41] and one with line assisted traction technique [42] were acquired from Prince of Wales Hospital for the purpose. The pocket creation technique is used to improve the visualization of dissection plane by creating a pocket in the submucosal layer after making a small mucosal incision for entry. On the other hand, traction method in ESD involves using additional instruments (such as a clip with a line, snare or other commercially available traction devices) to apply counter traction. Investigation into these techniques helps understand how the performance of AI model changes with different procedural steps, as well as tissue-tool interactions."*
- *"For validation on novel ESD techniques, AI-Endo maintains an average accuracy of 93.07% (CI: 83.44%, 100.0%) on pocket creation method. AI-Endo retains the ability to recognize surgical phases in the pocket creation process, even though the pocket creation is relatively new and not included in our developmental dataset. This advantage is largely attributed to its potential in capturing features of tissue background and tissue-tool interactions, which are shared between conventional operations and pocket creation. The accuracy on ESD with line-assisted traction was lower at 75.22% (CI is not calculated for one case). The*

reduction in accuracy was caused by the emergence of new functional tools (Figure 3d) during traction application. This limitation coincides with our expectation because it is considerably difficult to include all specialized tools and novel ESD techniques over time in our developmental dataset. Therefore, for the sack of knowledge on new scenes, AI-Endo generally predicted operations with new specialized tools as phase Idle. These experimental results are important to verify that the AI model trained on expert retrospective data is applicable to young generation of surgeons and novel techniques over time."

Reviewer Comment 2.7 — “Is AI-Endo only applicable for Olympus systems? How about Fujifilm and Pentax systems.”

Reply: In our study, the AI system was developed and validated based on data from Olympus system, making it most applicable to that system. However, during endoscopic view in ESD procedures, conventional white light images are used and these images remain largely consistent across different brands of endoscopes. Additionally, the design and implementation of intelligent algorithms in the development of AI-Endo did not depend on assumptions about the type of instrument being used. Therefore, it is believed that AI-Endo could be applied even in procedures using different endoscopy systems.

1. Application to different endoscopy systems

To verify the feasibility of AI-Endo in other systems, we collected four cases of ESD performed with the Fujifilm endoscopy system from Nanfang Hospital, Southern Medical University, Guangzhou, China. AI-Endo yielded an average accuracy of 90.75% (88.50%, 93.01%), which is comparable to the high performance it demonstrated on procedures using the Olympus system (91.04% (CI: 89.57%, 92.51%)). Practically, AI-Endo can accept the video stream and process data in a relatively independent manner, which means the inference speed should not be heavily dependent on the endoscopy system. As long as the video can be exported and read by a USB port, the proposed AI-Endo should be able to recognize surgical phases at around 47 fps, serving as a plug-in for real-time applications. Thus, the model can be successfully applied to systems other than Olympus.

2. Application to international multi-center datasets

In addition to validating the performance of AI-Endo on different endoscopy systems, we also explored its application to multinational external datasets. In addition to the four ESDs from Guangzhou, China, four more ESD videos were obtained from the Hospital of Augsburg's Internal Medicine III - Gastroenterology (Augsburg, Germany; Olympus), thus completing an international multi-center validation dataset. Finally, AI-Endo achieved an average accuracy of 90.75% (CI: 88.50%-93.01%) on cases from Guangzhou and 87.34% (CI: 84.43%-90.25%) on cases from Augsburg, which are comparable to the performance on the developmental dataset. The ROC curves for these two centers are shown in Figure 9a and Figure 9b, and phase-wise metrics for the two datasets are shown in Figure 9d. AI-Endo maintains high performance on all phases with most metrics greater than 85%. Moreover, we statistically analyzed the performance of AI-Endo on organs esophagus, colorectum, and stomach, on which AI-Endo still keeps a high

Figure 9: The application of AI-Endo to international multi-center datasets. (a) The ROC curve of AI-Endo on cases from Augsburg, Germany; (b) The ROC curve of AI-Endo on cases from Guangzhou, China; (c) Average accuracy on cases of esophagus, colorectum, and stomach; (d) Phase-wise performance metrics of AI-Endo on multi-center datasets. CI is not calculated for Marking phase in cases from center at Guangzhou, China because only one case involves marking.

level of accuracy (Figure 9c). These results suggest that AI-Endo relies not only on the tissue background features but also on the interaction between tissues and surgical tools, which remains relatively consistent in ESD surgery across different centers, contributing to the robustness of the system.

In the revised manuscript, we have added the section “*International multi-center external validation*” to discuss whether AI-Endo can be applied to other endoscopy systems other than Olympus and worldwide centers outside of Hong Kong (Page 8, lines 58-81; Page 9, lines 1-31).

“*Multi-center datasets provide a diverse range of samples, which are crucial for assessing the generalizability of deep learning models across different regions. This is especially vital in surgical applications, where patient populations and medical practitioners may differ in terms of lifestyle and regional training [47]. Additionally, with the advancements in surgical equipment, the endoscopy system may vary across different centers. It is essential to examine the performance of AI-Endo regarding different endoscopy systems. To this end, we collected data from multiple centers outside of Hong Kong, which showcased variance in both surgical practices and endoscopy systems compared to our developmental dataset. Through an analysis of AI-Endo’s robustness on multi-center datasets, we strive to ensure its applicability to a broader range of patients and surgical contexts, ultimately contributing to clinical practice.*

External validation on data from Augsburg, Germany. We obtained four ESD videos from Hospital of Augsburg’s Internal Medicine III - Gastroenterology, Augsburg, Germany. These videos were recorded using the same endoscopy system as the one utilized in our developmental dataset. Although the cases were conducted at international centers, AI-Endo maintained its

high performance and achieved an average accuracy of 87.34% (CI: 84.43%, 90.25%), specificity of 86.01% (CI: 71.48%, CI: 96.27%), and average sensitivity of 86.60% (CI: 74.21%, 96.36%). The ROC curve is illustrated in Figure 6a, with the derived metrics in Figure 6d. Both figures demonstrate the superiority of AI-Endo in overcoming the difference among countries. These findings suggest that AI-Endo can robustly extract phase-related that commonly exists in ESD procedures, such as tool features and tissue-tool interactions.

External validation on data from Guangzhou, China. As AI-Endo was developed using data solely from Olympus Endoscopy, it is crucial to assess its feasibility with other endoscopy systems. To achieve this, we acquired four cases from Nanfang Hospital, Southern Medical University in Guangzhou, China. Unlike the dataset previously discussed, the cases from Nanfang Hospital were imaged using Fujifilm Endoscopy system. All cases were annotated and processed in the same manner as the developmental dataset. AI-Endo finally yielded an average accuracy of 90.75% (CI: 88.50%, 93.01%) and exceptional ROC curves for each phase (Figure 6b). Furthermore, all phase-wise performance metrics were higher than 88% (Figure 6d), indicating exceptional performance comparable to that of the developmental dataset (Figure 2b). This investigation shows that AI-Endo's performance is robust and generalizable across different endoscopy systems.

During endoscopic procedures, conventional white light images are used and these images remain largely consistent across different brands of endoscopes. Additionally, the design and implementation of intelligent algorithms in the development of AI-Endo did not depend on assumptions about the type of instrument being used. AI-Endo can accept the video stream and process data in a relatively independent manner, which means the inference speed should not be heavily dependent on the endoscopy system. As long as the video can be exported and read from USB port, the proposed AI-Endo is able to recognize surgical phases at around 47 fps, serving as a plug-in for real-time applications. Therefore, it is believed that AI-Endo could be applied even in procedures using Fujifilm and Pentax endoscopes. Besides, we statistically analyzed the performance of AI-Endo on organs esophagus, colorectum, and stomach, on which AI-Endo keeps a high level of accuracy (Figure 6c). These experiments demonstrate the great potential of AI-Endo for worldwide applications across various organs, different centers, and multiple endoscopy platforms.”

Reviewer Comment 2.8 — “The performance would be changeable according to the GI organs (esophagus, stomach, duodenum, colon and rectum). Which organ is the main application of AI-Endo?”

Reply: We thank the reviewer for referring to the main application of AI-Endo. Because the formulation and implementation of phase recognition framework did not make specific assumptions about which kind of GI organ is used, the developed AI-Endo is supposed to be widely applicable to all GI organs.

For instance, in our study, we conducted an in-vivo animal experiment in which two surgeons conducted 12 cases at three different locations, colorectum, stomach, and esophagus. The accuracy rates of colorectum, stomach and esophagus are 83.29% (CI: 77.43%, 89.15%), 83.05% (CI:

78.11%, 87.99%) and 84.31% (CI: 78.77%, 89.85%) respectively, showing marginal differences. The exceptional generalization ability of AI-Endo across different organs is largely attributed to the robust features that commonly exist in ESD surgery, including but not limited to tissue-tool interactions and background tissues. Therefore, AI-Endo is capable of producing stable and reliable results in various GI organs.

In the revised manuscript, we have specified the generalization ability of AI-Endo to different GI organs at page 7, lines 13-17:

" Additionally, AI-Endo achieved accuracy rates of 83.29% (CI: 77.43%, 89.15%), 83.05% (CI: 78.11%, 87.99%) and 84.31% (CI: 78.77%, 89.85%) on colorectum, stomach and esophagus, respectively, showing marginal differences among different GI organs."

References

- [1] K. Hotta, T. Oyama, T. Shinohara, Y. Miyata, A. Takahashi, Y. Kitamura, and A. Tomori, "Learning curve for endoscopic submucosal dissection of large colorectal tumors," *Digestive Endoscopy*, vol. 22, no. 4, pp. 302–306, 2010.
- [2] I. Oda, T. Odagaki, H. Suzuki, S. Nonaka, and S. Yoshinaga, "Learning curve for endoscopic submucosal dissection of early gastric cancer based on trainee experience," *Digestive Endoscopy*, vol. 24, pp. 129–132, 2012.
- [3] Y.-K. Tsou, W.-Y. Chuang, C.-Y. Liu, K. Ohata, C.-H. Lin, M.-S. Lee, H.-T. Cheng, and C.-T. Chiu, "Learning curve for endoscopic submucosal dissection of esophageal neoplasms," *Diseases of the Esophagus*, vol. 29, no. 6, pp. 544–550, 2016.
- [4] H. S. Choi and H. J. Chun, "Accessory devices frequently used for endoscopic submucosal dissection," *Clinical endoscopy*, vol. 50, no. 3, pp. 224–233, 2017.
- [5] Q. Dou, H. Chen, L. Yu, L. Zhao, J. Qin, D. Wang, V. C. Mok, L. Shi, and P.-A. Heng, "Automatic detection of cerebral microbleeds from mr images via 3d convolutional neural networks," *IEEE transactions on medical imaging*, vol. 35, no. 5, pp. 1182–1195, 2016.
- [6] S. Ebrahimi Kahou, V. Michalski, K. Konda, R. Memisevic, and C. Pal, "Recurrent neural networks for emotion recognition in video," in *Proceedings of the 2015 ACM on international conference on multimodal interaction*, 2015, pp. 467–474.
- [7] A. v. d. Oord, S. Dieleman, H. Zen, K. Simonyan, O. Vinyals, A. Graves, N. Kalchbrenner, A. Senior, and K. Kavukcuoglu, "Wavenet: A generative model for raw audio," *arXiv preprint arXiv:1609.03499*, 2016.
- [8] D. J. Hand and R. J. Till, "A simple generalisation of the area under the roc curve for multiple class classification problems," *Machine learning*, vol. 45, pp. 171–186, 2001.

- [9] M. De Figueiredo, C. B. Cordella, D. J.-R. Bouveresse, X. Archer, J.-M. Bégué, and D. N. Rutledge, “A variable selection method for multiclass classification problems using two-class roc analysis,” *Chemometrics and Intelligent Laboratory Systems*, vol. 177, pp. 35–46, 2018.
- [10] C. T. Nakas, T. A. Alonzo, and C. T. Yiannoutsos, “Accuracy and cut-off point selection in three-class classification problems using a generalization of the youden index,” *Statistics in medicine*, vol. 29, no. 28, pp. 2946–2955, 2010.
- [11] J. Martin, G. Regehr, R. Reznick, H. Macrae, J. Murnaghan, C. Hutchison, and M. Brown, “Objective structured assessment of technical skill (osats) for surgical residents,” *British journal of surgery*, vol. 84, no. 2, pp. 273–278, 1997.
- [12] J. D. Doyle, E. M. Webber, and R. S. Sidhu, “A universal global rating scale for the evaluation of technical skills in the operating room,” *The American journal of surgery*, vol. 193, no. 5, pp. 551–555, 2007.
- [13] P. Mascagni, D. Alapatt, L. Sestini, M. S. Altieri, A. Madani, Y. Watanabe, A. Alseidi, J. A. Redan, S. Alfieri, G. Costamagna *et al.*, “Computer vision in surgery: from potential to clinical value,” *npj Digital Medicine*, vol. 5, no. 1, p. 163, 2022.
- [14] T. Lambin, J. Rivory, T. Wallenhorst, R. Legros, F. Monzy, J. Jacques, and M. Pioche, “Endoscopic submucosal dissection: How to be more efficient?” *Endoscopy International Open*, vol. 9, no. 11, pp. E1720–E1730, 2021.
- [15] S. Hua, J. Gao, Z. Wang, P. Yeerkenbieke, J. Li, J. Wang, G. He, J. Jiang, Y. Lu, Q. Yu *et al.*, “Automatic bleeding detection in laparoscopic surgery based on a faster region-based convolutional neural network,” *Annals of Translational Medicine*, vol. 10, no. 10, 2022.
- [16] T. Okamoto, T. Ohnishi, H. Kawahira, O. Dergachyava, P. Jannin, and H. Haneishi, “Real-time identification of blood regions for hemostasis support in laparoscopic surgery,” *Signal, Image and Video Processing*, vol. 13, pp. 405–412, 2019.
- [17] M. Kitamura, Y. Miura, S. Shinozaki, A. K. Lefor, and H. Yamamoto, “The pocket-creation method facilitates endoscopic submucosal dissection of gastric neoplasms along the lesser curvature at the gastric angle,” *Frontiers in Medicine*, vol. 9, p. 825325, 2022.
- [18] M. Yoshida, K. Takizawa, S. Suzuki, Y. Koike, S. Nonaka, Y. Yamasaki, T. Minagawa, C. Sato, C. Takeuchi, K. Watanabe *et al.*, “Conventional versus traction-assisted endoscopic submucosal dissection for gastric neoplasms: a multicenter, randomized controlled trial (with video),” *Gastrointestinal Endoscopy*, vol. 87, no. 5, pp. 1231–1240, 2018.

END

REVIEWERS' COMMENTS

Reviewer #1 (Remarks to the Author):

Thanks you for being responsive to the first review. The current paper is an improved version, my open questions and concerns were carefully addressed including clarification of missing details, additional experiments including a multi-center evaluation, a better description of the results as well as the promise to publish parts of the data. One minor issue: please include in the discussion a comment regarding the performance decrease on specific phases regarding the ex-vivo/in-vivo dataset.

Reviewer #2 (Remarks to the Author):

I carefully reviewed the revised manuscript. I think the contents are improved with satisfaction. I have no further comments and questions. Perhaps, "and randomized clinical trials," would be "at randomized clinical trials," in the abstract and " Normalized Transaction index" would be "Normalized Transition index" in Figure 5a.

Response letter for manuscript NCOMMS-23-02595A

Intelligent Surgical Workflow Recognition for Endoscopic Submucosal Dissection with Real-time Animal Study

We sincerely thank the reviewers again for their recognition of this work and their valuable feedback during the major revision process. In this final version, we have addressed the new comments from reviewers point-by-point as follows.

Response to Reviewer 1

Comment: *“Thank you for being responsive to the first review. The current paper is an improved version, my open questions and concerns were carefully addressed including clarification of missing details, additional experiments including a multi-center evaluation, a better description of the results as well as the promise to publish parts of the data. One minor issue: please include in the discussion a comment regarding the performance decrease on specific phases regarding the ex-vivo/in-vivo dataset.”*

Response: We greatly appreciate the positive comments from the reviewer. To discuss about the performance of AI-Endo on ex-vivo and in-vivo dataset, we have included the comment on performance decline of phase marking at Page 9, Lines 108-122.

“Second limitation of this work concerns the model generalizability, which was noticeable from the performance drop observed in ex/in-vivo animal experiments (Supplementary Tables 3 and 4). Despite this would be explained by appearance difference between animal tissue and human tissue, similar degradation is anticipated to encounter under the emergence of new tools (Figure 3d) that were not covered by the developmental data. Relatively small-size dataset limits the model's robustness to identify effective tool features or surgical scenes when unseen ESD technique is involved. Our currently developed method has not particularly addressed this problem, while can be extendable with domain generalization [48] and test-time adaptation [49] strategies. Promisingly, the proposed model has shown a noteworthy degree of adaptability to the variations encountered in surgical settings, such as differences in geographical locations and endoscopy systems. This matters its wider application and multi-center deployment in the future.”

Response to Reviewer 2

"I carefully reviewed the revised manuscript. I think the contents are improved with satisfaction. I have no further comments and questions. Perhaps, "and randomized clinical trials," would be "at randomized clinical trials," in the abstract and "Normalized Transaction index" would be "Normalized Transition index" in Figure 5a."

Response: We thank the reviewer for pointing out the typos. We have changed these typos accordingly.